

# Bedload transport fluctuations, flow conditions and disequilibrium ratio at the Swiss Erlenbach stream: results from 27 years of high-resolution temporal measurements

Dieter Rickenmann[1]

[1]Swiss Federal Research Institute WSL, Birmensdorf, 8903, Switzerland

*Correspondence to*: Dieter Rickenmann (dieter.rickenmann@wsl.ch)

**Abstract.** Based on measurements with the Swiss plate Geophone system with a 1 min temporal resolution, bedload transport fluctuations were analysed as a function of the flow and transport conditions in the Swiss Erlenbach stream. The study confirms a finding from an earlier event-based analysis of the same bedload transport data, which showed that the disequilibrium ratio of measured to calculated transport rate influences the sediment transport behaviour. To analyse the transport conditions, the following elements were examined to characterise bedload transport fluctuations: (i) the autocorrelation coefficient of bedload transport rates as a function of lag time, (ii) the critical discharge at the begin and end of a transport event, (iii) the coefficient of variation of the bedload transport rates, and (iv) a hysteresis index as a measure of the strength of clockwise or anticlockwise transport behaviour. This study underlines that above average disequilibrium conditions, which are associated with a larger sediment availability on the streambed, have generally a stronger effect on subsequent transport conditions than below average disequilibrium conditions, which are associated with comparatively less sediment availability on the streambed. The findings highlight the important roles of the sediment availability on the streambed, the disequilibrium ratio, and the hydraulic forcing in view of a better understanding of the bedload transport fluctuations in a steep mountain stream.

## 1 Introduction

A recent comprehensive review of bedload transport fluctuations was compiled by Ancey (2020a, 2020b). He concluded that predictions of transport levels from bedload transport relationships are typically associated with an uncertainty of (at least) about one order of magnitude, and that morphodynamic models fail to explain bedform evolution without the use of additional assumptions (Ancey, 2020a). He also noted that bedload transport rates depend on many processes that vary in time and space, and are interrelated (Ancey, 2020b). Some aspects influencing the fluctuations of bedload transport, apart from the primary control by hydraulic forcing, are summarised below, focusing on those that are relevant for this study.

Possible reasons for the variability of bedload transport rates were previously discussed in Rickenmann (2020). One reason is related to changes in bed evolution as a result of bedload transport affecting the critical Shields stress necessary for the start of bedload transport, such as changes in packing arrangement, grain sizes of the bed surface, roughness, imbrication and





orientation of the particles, cluster structure, and armor layer configuration (e.g., Church, 2006; Piedra et al., 2012; Mao, 2012; Guney et al., 2013; Roth et al., 2017). A second reason is the stabilization of the bed surface during low-flow periods without bedload transport (inter-event flows) and during flow periods with only weak transport, leading to an increase in critical Shields stress or a reduction in sediment transport (e.g., Paphitis and Collins, 2005; Haynes and Pender, 2007; Ockelford et al., 2019). Such a stabilization of the bed surface was also confirmed for the Swiss Erlenbach stream (Masteller

et al., 2019). A third reason for the transport variability is sediment availability on the bed surface, caused either by high discharge events breaking up channel bed structures (Turowski et al., 2009; Yager et al., 2012), or by higher rates of upstream sediment supply resulting both in more mobile sediment and thus in lower critical Shields stresses (Dietrich et al., 1989; Recking, 2012; Bunte et al., 2013; Rickenmann, 2020). Based on flume experiments, An et al. (2021) examined the effect of antecedent conditioning flows with different durations (stress history) followed by a hydrograph with increasing

discharge (only rising limb) and sediment input on changes in bed elevation, surface grain size and bedload transport. They concluded that the effect of stress history on sediment transport rate is important at the beginning of a hydrograph and diminishes with the increase of discharge and sediment supply, suggesting a loss of memory of stress history under high flow conditions.

The correlation between bedload transport and water discharge was found to clearly increase when averaging measurements

of bedload transport over increasing time periods (Downs et al., 2016; Lenzi et al., 2004; Recking et al., 2012; Rickenmann, 1994, 2016, 2018; Rickenmann and McArdell, 2008). Based on continuous bedload transport measurements with the Swiss Plate Geophone (SPG) system, Rickenmann (2018) reported a substantial increase in the correlation coefficient R between transport rate and discharge for minimum aggregation times of about 1–2 h, which integrates over typical short-term fluctuations of 15–35 min of bedload transport rates in natural gravel bed streams. Such short-term fluctuations were found

by assessing the temporal variability of bedload transport at a time scale of less than 1 h for the Drau River in Austria by Habersack et al. (2012) and for the Elwha River in the U.S. by Hilldale (2015). In both studies, SPG measurements were used with a 1 min recording interval of geophone impulses, which showed a periodicity of bedload peaks ranging from 15 to 35 min, based on moving 5 min average values. Fluctuations with a periodicity of 14 to 35 min were found also in a field study of the Turkey brook in England (Reid and Frostick, 1986) and in flume experiments (Kuhnle and Southard, 1988,

Strom et al., 2004) for a variety of hydraulic and sediment supply conditions. Moving bed load sheets or the formation and destruction of gravel clusters were suggested as possible reasons for such fluctuations (Hilldale, 2015; Kuhnle and Southard, 1988).

Liébault et al. (2022) investigated the behaviour of seasonal bedload pulses in a small alpine catchment, and they concluded that the mean bedload response of the alluvial system is strongly controlled by sediment storage within the channel system,

as evidenced by larger bedload fluxes at the catchment outlet during degradational phases in the channel upstream. Brenna and Surian (2023) concluded from a field study that fluxes of coarse sediment in a mountain stream in Northern Italy recently affected by a large flood could be considerably higher than those normally expected, because of the high availability of fresh and unstructured sediment within the channel. A similar explanation of such a memory effect was proposed by



Turowski et al. (2009), Yager et al. (2012), and Masteller et al. (2019), who observed that extreme flood events in mountain
streams can destabilise the channel bed and increase sediment availability and following fluxes.

Autocorrelation analysis was used to assess memory effects in bedload transport time series. Elgueta Astaburuaga et al.
(2018) analysed the effect of sediment supply on sediment mobility for a poorly sorted experimental bed, and they suggested
that large sediment pulses may increase the strength and persistence of autocorrelation in bedload rate time series. In another
flume study, Saletti et al. (2015) found from autocorrelation analysis that memory is grain-size dependent, highlighting the
importance of fractional transport data for an accurate description of bed load dynamics. Masteller et al. (2919) showed for
field data from the Erlenbach stream that the critical Shields stress at the start of an event remained significantly
autocorrelated over up to about ten transport events. Rickenmann (2020) found a similar memory effect for the Erlenbach for
critical Shields stress at the end of an event and for the disequilibrium ratio.

Continuous and longer time series of bedload transport measurements are still scarce for field situations. Therefore, the
majority of studies on fluctuations of bedload transport rely on flume experiments. Mettra (2014) investigated bedload
transport fluctuations with a series of flume experiments and considered the effect of varying sediment input at the upstream
end of the flume. For increasing values of unit bedload transport rates $q_b$ at the flume outlet, varying from about 0.33 to 600
g s$^{-1}$ m$^{-1}$, he found an almost linear decrease in the coefficient of variation ($cv$) from about 10 to 0.1. He also observed that
intermittency of transport increased with increasing channel steepness, decreasing sediment supply and decreasing bulk flow
energy. Furthermore, he found that lower sediment supply, higher channel slope and lower Shields stress lead to more
intense hysteretic effects. Singh et al. (2009) performed flume experiments with $q_b$ = 0.36 and 730 g s$^{-1}$ m$^{-1}$ and determined
$cv$ values of 0.98 and 0.76 for aggregation times of 1 min. From their flume experiments, Kuhnle and Southard (1988)
determined for 1 min sampling times for $q_b$ = 52 to 541 g s$^{-1}$ m$^{-1}$ $cv$ values from 0.33 to 0.5, and for $q_b$ = 6000 g s$^{-1}$ m$^{-1}$ a $cv$
value of 0.13. Ma et al. (2014) performed flume experiments with spherical beads, and for an integration time of 1 min they
measured a mean bedload flux of 1.1 particle/s and determined a $cv$ value of 1.29.

Regarding direct bedload measurements, Kuhnle and Willis (1998) used data from Goodwin creek in the USA and
determined for $q_b$ = 7.4 to 550 g s$^{-1}$ m$^{-1}$ and a 1 min sampling time $cv$ values from 2.42 to 0.78, with the bulk of values
varying between 1 and 2; he also found a tendency for the $cv$ values to decrease with increasing bedload transport rates.
Other field studies cited below were based on impact plate measurements with the SPG system. For the Elwha River in the
USA, Hilldale (2015) analysed impulse counts with a recording interval of 1 min for two different events and found a $cv$
value of 0.92 and of 0.78, respectively, for a unit flow discharge of about 2.1 m$^3$ s$^{-1}$ m$^{-1}$. Ancey and Pascal (2020) analysed
bedload transport measurements in the Navisence River in Switzerland, and for an approximately constant unit flow
discharge of about 1.2 m$^3$ s$^{-1}$ m$^{-1}$ over 1 hour and a mean value of $q_b$ = 39 g s$^{-1}$ m$^{-1}$ the resulting $cv$ value for a 1 min sampling
time was 0.41.

A comprehensive review of the hysteresis behaviour of sediment transport rates in alluvial streams was made by Gunsolus
and Binns (2018) who concluded that lower magnitude hydrographs resulted in more pronounced hysteresis than larger
magnitude hydrographs. Clockwise hysteresis (with higher transport rates on the rising than on the falling hydrograph limb)



has often been associated with a gradual decrease in sediment availability or early exhaustion of sediment sources (Pretzlav et al., 2020; Mao et al., 2019; Rovira and Batalla 2006; Gao and Pasternack, 2007). Anticlockwise hysteresis was suggested

to result from temporal lags as bedforms adjust to changing discharge (Bombar et al., 2011; Martin and Jerolmack, 2013) or from the destabilization of surface structures during hydrograph rising limbs (Kuhnle, 1992).

The objective of this study is to examine the bedload transport fluctuations as a function of the flow and transport conditions in the Swiss Erlenbach stream. The study is based on 27 years of 1 min time series of bedload transport rates, measured with the SPG system, for the same 522 flood events which were used previously for the analysis of event-based transport

characteristics (Rickenmann, 2020). The disequilibrium ratio was determined as the ratio of measured to calculated transport rate, similarly, as defined in Rickenmann (2020). This new study focuses on the following elements that characterise bedload transport fluctuations: (i) the autocorrelation coefficient of bedload transport rates as a function of lag time, (ii) the critical discharge at the begin and end of a transport event, (iii) the coefficient of variation of the bedload transport rates, and (iv) a hysteresis index as a measure of the strength of clockwise or anticlockwise transport behaviour. These elements are analysed

and discussed in the context of variations in discharge, bedload transport level, and disequilibrium ratio. The findings highlight the important roles of the sediment availability on the streambed, the disequilibrium ratio, and the hydraulic forcing in view of a better understanding of the bedload transport fluctuations in a steep mountain stream.

## 2 Field site, measurements, and data analysis

### 2.1 The Erlenbach catchment

The Erlenbach catchment is situated in the Alptal valley in the Pre-alps of central Switzerland and has an area of ~0.7 km$^2$ (Rickenmann et al., 2012). Geologically, the Erlenbach basin is located in a Flysch zone; creeping and sliding slopes along most of the channel length provide a persistent high supply of sediment to the channel (Schuerch et al., 2006; Golly et al., 2017). The hydrology of the Erlenbach catchment is characterised by both frequent high intensity storms in summer with discharge events of short duration and some snowmelt events in spring. Annual precipitation totals are around 2300 mm

(Rickenmann and McArdell, 2007).

The stream gradient is 18% on average and 10.5% along the 50 m long natural reach immediately upstream of the gauging site (Fig. 1). The Erlenbach has a pronounced step–pool–riffle morphology (Rickenmann and McArdell, 2007; Turowski et al., 2009). The channel is situated in alluvial materials, originating from weathered Flysch bedrock, with grain sizes ranging from clay to boulders. Most of the catchment is on a large landslide complex, and the left bank is particularly active in a

reach 500 m upstream of the gauging station (Schuerch et al., 2006). In this reach, the steps - composed of both coarse particles and large wood - are up to 2 m high, with a mean step height of 0.7 m and a mean step spacing of almost 8 m (Molnar et al., 2010). Bedload transport events occur on average about 20 times per year. The three largest sediment transport events had peak discharges of about 10 m$^3$ s$^{-1}$ or more and resulted in transported sediment volumes (including fine material) of more than 1000 m$^3$ (Rickenmann, 2020).



## 2.2 Bedload transport measurements

Near the catchment outlet there is a sediment retention basin (Fig. 1a). Downstream of the lowest natural channel reach, an artificial channel with embedded riprap leads to a large check dam, forming the upstream end of this retention basin. Since 1986 several adjacent steel plates with impact sensors are embedded in the check dam to continuously measure bedload transport. These sensors record the vibrations resulting from the movement of gravel sized and larger particles over a steel plate (Rickenmann and McArdell, 2007). From 1986 to 1999 piezoelectric impact sensors (PBIS), developed in-house, were used. Because of deterioration of these PBI sensors, they were replaced by geophone sensors from 2000 onwards (Rickenmann, 2017). The bedload measurements with both sensors result in a very similar signal response (Rickenmann et al., 2012). The minimum grain size that can be detected by both systems is about 10 mm (Rickenmann and McArdell, 2007; Rickenmann et al., 2012; Rickenmann et al., 2014; Wyss et al., 2016a, 2016b; Nicollier et al., 2022). A vibration sensor is fixed from underneath to the center of a steel plate. The steel plates have standard dimensions of $L \times B \times T = 358 \times 496 \times 15$ mm, where L is the downstream length, B is the transversal width, and T is the thickness of the plate.

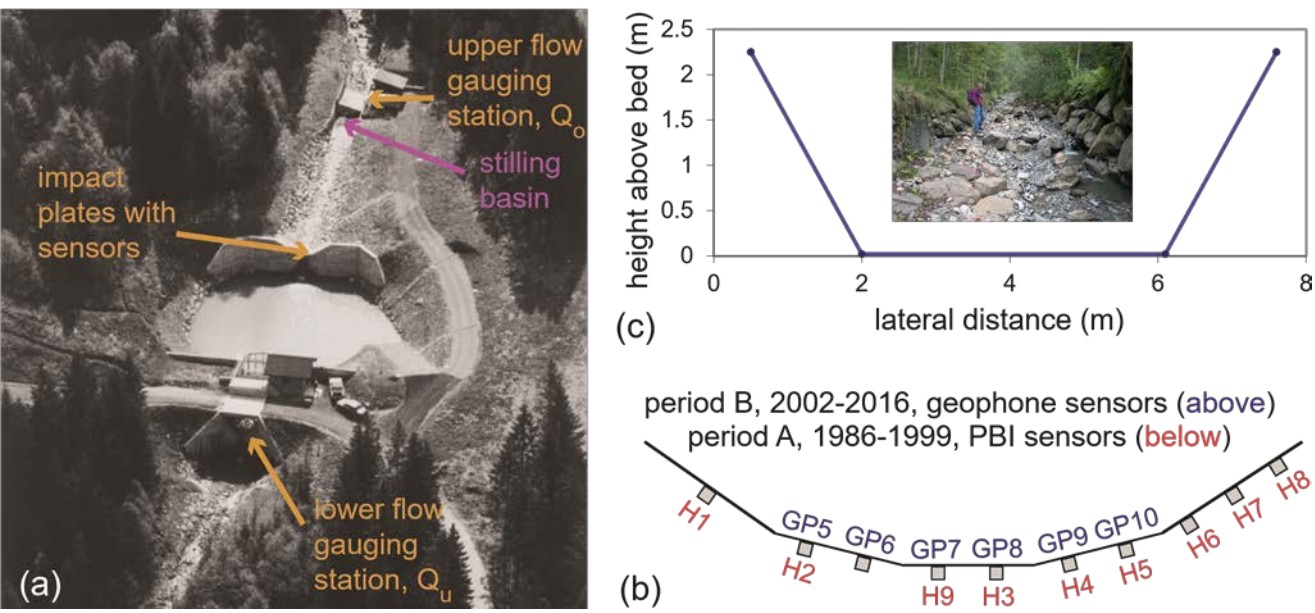

**Figure 1. (a) Surrogate bedload impact plates, sediment retention basin, and two gauging sites at the Erlenbach stream. (b) Schematic cross-section, with downstream view of the lateral distribution of PBI sensors (Hi) during period A (1986-1999) and of geophone sensors (GPi) during period B (2002-2016), where i denotes the sensor number. (c) Symmetrical, trapezoidal cross-section used for the hydraulic and bedload-transport calculations, with a bottom width $b_w = 4.1$ m and a lateral slope of 1.5:1; this cross-section represents the natural reach upstream of the measuring installations, with the banks protected by a riprap construction (inset photo, upstream view). (Modified from Rickenmann, 2020)**

For the calibration of the surrogate measuring system, the summary value of the impulse counts is used in this study: whenever the voltage of the raw signal exceeds a preselected threshold value $A_t$ (in V), this is recorded as an impulse and the





summed impulses are stored. The threshold value to record impulses was set to $A_t = 0.2$ V for the PBI and to $A_t = 0.1$ V for the geophone sensors. During bedload-transporting flood events, the summed impulse counts $IMP$ were recorded in minute intervals. The use of a threshold essentially eliminates the noise of the signal. A linear relation between impulses and bedload mass transported over a plate was found to apply quite well at several field sites, including the Erlenbach

(Rickenmann et al., 2012, 2014, 2020; Wyss et al., 2016a, 2016c; Nicollier et al., 2021). For the period A with the PBI sensors (1986-1999), a total of 9 plates were equipped with a sensor, and the lateral distribution of these sensors is illustrated in Fig. 1b. There is an asymmetrical lateral distribution of bedload transport intensities, due to an offset of the centerline axes of the approach flow channel with respect to the position of the check dam upstream of the retention basin. As a result, most signal was recorded by sensor H3, followed by sensors H4 and H9 (Rickenmann and McArdell, 2007). For the period B with

the geophone sensors (2002-2016), a total of 6 plates were equipped with a sensor, and the lateral distribution of the devices is illustrated in Fig. 1b. A similar asymmetrical lateral distribution of bedload transport intensities was evident also for this period.

The linear calibration relation for the impact plate measurements used to convert the number of impulses into a channel-wide bedload transport rate $Q_b$, including grains larger than 4 mm for both periods was determined as:

$\quad Q_b$ (kg s$^{-1}$) $= k_{isp}$ ($IMP$/1000) $m_{fg}$ $\rho_d$ $f_c$ / 60 $= k_{isp}$ ($IMP$/1000) $f_g$ $\hspace{3cm}$ (1)

where $k_{isp}$ are the individual calibration coefficients determined for each survey interval of the deposits in the sediment retention basin, $\rho_d = 1750$ kg m$^{-3}$ is the mean bulk density of the deposits (Rickenmann and McArdell, 2007), the coefficient $f_c = 0.5$ accounts for an estimated 50% of particles having a grain size larger than 10 mm (Rickenmann and McArdell, 2007), and the coefficient 60 is used to convert the 1 min $IMP$ readings to a mean value of $Q_b$ expressed in (kg s$^{-1}$). Finally, the

coefficient $m_{fg}$ is used to account for the fraction of particles in the range of "fine gravel", i.e. for 4 mm $< D <$ 10 mm, where $D$ is particle size. The value of $m_{fg}$ was estimated with the aid of the Erlenbach moving basket system to collect bedload samples (Rickenmann et al., 2012). The standard mesh size of the metal basket is 10 mm. For 11 samples from 2013 to 2017 a second layer of a metal wire net with a mesh size of 2 mm was inserted into one of the three baskets. From these samples a mean value of $m_{fg} = 1.54$ was determined, to correct for non-measured particles in the range 4 mm $< D <$ 10 mm with the

$IMP$ counts. The individual $m_{fg}$ values showed a slight decrease with increasing transport rate $Q_b$, whereby $Q_b$ was smaller than 1 kg s$^{-1}$ for all 11 samples. Thus, the lumped factor $f_g = 22.46$ in Eq. (1) accounts for the mean bulk density of the deposits in the sediment retention basin, the estimation of the proportion of particles larger than 2 mm in the deposits, and for the conversion of the 1 min $IMP$ readings to a mean value of $Q_b$ expressed in (kg s$^{-1}$). The originally measured $Q_b$ also include zero values and are termed $Q_{bz}$ henceforth (for comparison with a full time series $Q_{bM}$ for which zero values were

replaced with non-zero values, see section 2.6).

For period A, only the recordings with sensor plate H3 (Fig. 1b) were used to determine the individual calibration coefficients $k_{isp}$, because data from other sensors were sometimes missing. For period B, the sum of the impulses from sensor plates GP6, GP7, GP8, GP9, and GP10 (Fig. 1b) were used to determine the individual calibration coefficients $k_{isp}$; data from



the sensor plate GP5 were not included because this sensor malfunctioned for some later years. The resulting calibration
coefficients $k_{isp}$ are reported in the Supporting Information (Table S1) of Rickenmann (2020).

**2.3 Discharge measurements**

There are two discharge gauging stations near the sediment retention basin, which is close (~100 m) to the outlet of the
catchment. The so-called lower station at the outflow from the (dammed) sediment retention basin consists of a double
triangular profile; the associated stage-discharge relationship (discharge $Q_u$) was calibrated with a physical model before
construction and is therefore quite reliable also during flood flows. The upper discharge measurement station is located
between the end of the natural channel and the retention basin, and it consists of an asymmetrical cross-section in a concrete
channel; the associated stage-discharge relationship (discharge $Q_o$) was calibrated based on dye and salt tracer
measurements, especially for small and medium discharges, but including values up to 5 m$^3$ s$^{-1}$. The location of the two
gauging sites is illustrated in Fig. 1a and in Beer et al. (2015, Fig. 2 therein).

For the sediment-transporting flood events, discharges were in the range of 0.1 m$^3$ s$^{-1}$ to 15 m$^3$ s$^{-1}$ (Rickenmann, 1997;
Turowski et al., 2009). The definition of a sediment-transporting flood event is based on the recording of the bedload
transport activities. During such events, level measurements are available with a recording interval of 1 min for both stream
gauging sites. More details on the bedload transport criteria to start and end the recording of an event are given in the
Supporting Information for Rickenmann (2020, Text S2 therein).

Generally, I used the $Q_o$ values from the upper gauging station in this study. Due to uncertainty of the level-discharge
relationship for some (limited) periods, the $Q_o$ values from the upper gauging station were replaced by $Q_u$ values from the
lower gauging station. I estimated the uncertainty in the $Q_o$ values to be 15% or slightly less, similarly as Beer et al. (2015).
More details on the discharge measurements are given in the Supporting Information (Text S3) of Rickenmann (2020). For
simplicity, measured discharges used in the further analysis are labelled as $Q$ values in the following text.

Immediately downstream of the upper gauging site, the flow drops over an engineered over-fall structure (~1 m high) into a
shallow stilling basin (~4 m × 4 m × 0.3 m depth), then enters a fairly smooth ~30 m engineered concrete reach with large
blocks embedded in cement before reaching the impact plates (Fig. 1a; see also Roth et al., 2017, Fig. 1 therein). This stilling
basin slightly delays the sediment transfer between the (upper) gauging station and the impact plates.

**2.4 Sediment-transporting flood events**

The delineation of sediment-transporting flood events is described in detail in Rickenmann (2020). On some days with
bedload transport, the *IMP* counts ceased for minutes to hours, and restarted again. A minimum inter-flood duration time of
75 min (without any transport signal) was used to separate different transport events, which is of the order of the duration of
a typical flood event (see below). Furthermore, the smallest events were excluded from the further analysis due to
measurement uncertainty in detecting weak bedload transport. This resulted in 286 flood events for period A and in 236
210  flood events for period B (Rickenmann, 2020). For the total of 522 flood events, this delineation also allowed to determine



the threshold discharge at the start of an event, $Q_s$ (in m$^3$ s$^{-1}$), the threshold discharge at the end of an event, $Q_e$ (in m$^3$ s$^{-1}$), and the peak discharge observed during an event, $Q_{max}$ (in m$^3$ s$^{-1}$). The typical duration of the thunderstorm-triggered events during summer varied between about 20 min and 300 min, with a median duration of 94.5 min for period A and of 99.5 min for period B (Fig. S1).

## 2.5 Hydraulic and bedload-transport calculations

The hydraulic and the bedload-transport calculations were made as partly described in Rickenmann (2020). They were based on the measured trapezoidal cross-section in the natural reach upstream of the measuring installations close to the retention basin (Fig. 1c), using the mean channel bed slope of $S = 10.5\%$, and characteristic grain sizes for the Erlenbach bed surface material of $D_{84} = 0.29$ m and $D_{50} = 0.06$ m (Rickenmann, 2020). $D_{xx}$ is the grain size for which xx% of the particles are finer. The hydraulic calculations were made with a hydraulic geometry flow resistance relation developed by Rickenmann and Recking (2011), which was verified with field measurements in the Erlenbach by Nitsche et al. (2012). The bedload-transport calculations were performed with two equations reported in Schneider et al. (2015), which represent a modified form of the Wilcock and Crowe (2003) equation, and which predict total bedload transport rates (not fractional transport rates) for grain sizes larger than 4 mm. The first bedload-transport equation (SEA1) is based on total shear stress and a slope-dependent reference shear stress. The resulting calculated bedload transport rate for the entire channel width is $Q_{btot}$. The second bedload-transport equation (SEA2) is based on a reduced (effective) shear stress and a constant, slope-independent reference shear stress. The resulting calculated bedload transport rate for the entire channel width is $Q_{bred}$. Further details of both the hydraulic and the bedload-transport calculations are given in Appendix A1.

For the event-based analysis of the Erlenbach bedload transport measurements and using the SEA1 equation, Rickenmann (2020, Fig. A1 therein) found a reasonable correlation between the observed event bedload mass and the calculated event bedload mass, with a squared correlation coefficient $R^2 = 0.70$ (using log values), and a tendency to underestimate the calculated masses for the smaller events. The result of applying the hydraulic and the bedload-transport calculations to the 1 min values is shown in Fig. 2. There we can observe a similar underestimation of calculated bedload transport rates $Q_b$ for the smaller discharges (for both $Q_{btot}$ and $Q_{bred}$), and particularly for period B (2002 – 2016). However, the bedload-transport equations SEA1 and SEA2 agree reasonably well with the binned means of the measured bedload transport rates for discharges $Q$ larger than about 0.3 to 0.6 m$^3$ s$^{-1}$.





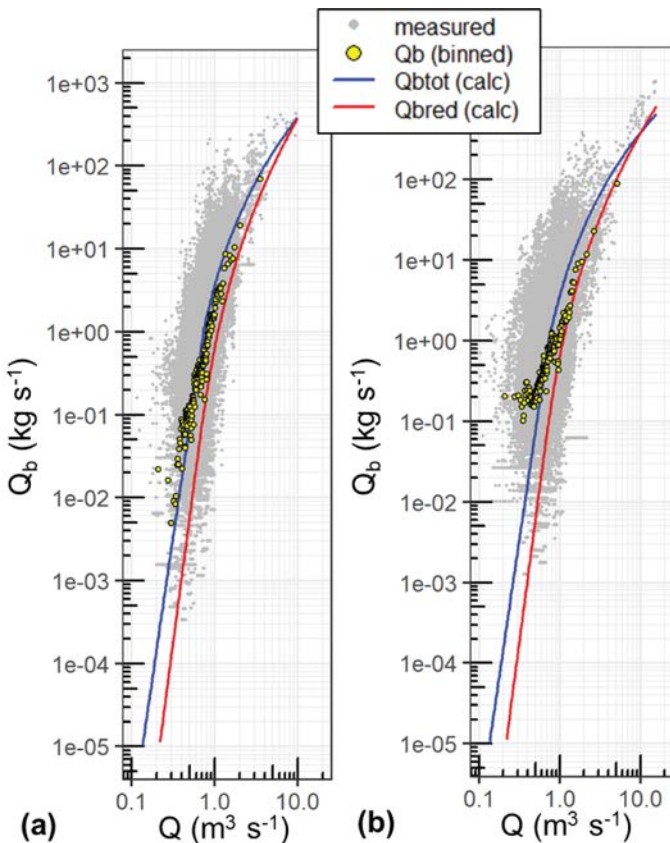

**Figure 2. Measurements of bedload transport $Q_b$ vs. discharge $Q$. $Q_b$ values are based (a) on the PBI sensors for the period A (1986 – 1999, grey dots), and (b) on the geophone sensors for the period B (2002 – 2016, grey dots). The binned values of $Q_b$ were calculated as the geometric mean, whereby zero values were replaced by the $Q_{bM}$ values (see section 2.6 for explanation). The red and blue lines represent the calculated bedload transport rates (described in section 2.5).**

## 2.6 Accounting for zero bedload transport values and use of the transport disequilibrium ratio

Observations of bedload transport often show a large variability for a given discharge value, typically covering several

orders of magnitudes, as illustrated in Fig. 2. In such a situation, when binning the measurements in discharge classes to determine a mean trend of the observed transport rates, it is preferable to calculate a geometric mean of the $Q_b$ values. However, this brings about the problem of how to deal with observed zero $Q_b$ values. Here, I took an approach similar to that proposed by Gaeuman et al. (2009, 2015), averaging the zero $Q_b$ values with (temporally) neighboring non-zero $Q_b$ values. I replaced the zero $Q_{bz}$ values by averaging any $k$ successive zero $Q_{bz}$ values by including the two neighboring non-zero $Q_{bz\_p}$

and $Q_{bz\_a}$ values, where the indices stand for p=prior and a=after a series of $k$ successive zero $Q_{bz}$ values. Then I assigned to all $(k+2)$ values the average value $Q_{bM} = (Q_{bz\_p} + Q_{bz\_a})/(k+2)$. The grey dots in Fig. 2 show the $Q_b = Q_{bM}$ values, and the binned $Q_b$ values in Fig. 2 were calculated as the geometric mean of the $Q_{bM}$ values for any given discharge class.



For the event-based analysis of bedload transport at the Erlenbach, I had used the disequilibrium ratio $E_d$, calculated as $M_{gravel}/M_{greg}$, where $M_{gravel}$ is the transported bedload mass per event, and $M_{greg}$ is the estimated bedload mass per event based

on the calculation of $Q_{btot}$ (Rickenmann, 2020). Here, I introduce a similar disequilibrium ratio $E_{dM}$, which is based on the measured 1 min values $Q_{bM}$ and on the calculated bedload transport rates $Q_{btot}$ (Eq. 2). The use of $Q_{bM}$ values has the advantage to facilitate the analysis using the full data set including all minute values.

$$E_{dM} = Q_{bM} / Q_{btot} \tag{2}$$

One goal of the present study was to compare some general results of the analysis using the 1 min values with the event-

based analysis reported in Rickenmann (2020), such as memory effects and the negative correlation between threshold discharge and disequilibrium ratio. There, it was observed that there was a memory effect for the $E_d$ values, and also for the $Q_{b,s}$ and the $Q_{b,e}$ values for a lag of at least three events for period A and somewhat longer for period B. It is also known from earlier studies that the correlation between $Q_b$ and $Q$ increases for aggregation times up to 60 min (Rickenmann, 2018). Given a typical event duration of 60 to 90 min, and that at least 85 % of the events had durations of more than 30 min (Fig.

S1), the following smoothening procedure was applied to the 1 min values, in order to delineate typical cycles with $E_{dM}$ values above and below the median for all events of a given period: (i) a kernel smoothing was applied to the $Q_{bM}$ values with a bandwidth of 30, resulting in the $Q_{bks30}$ values; (ii) smoothened $E_{dM}$ values were first determined as $E_{dks30} = Q_{bks30}/Q_{btot}$; (iii) then a kernel smoothing was applied to the log($E_{dks30}$) values with a bandwidth of 300, resulting in the so-called $E_{dks}$ values. This last step represents roughly a smoothening of the $E_{dks30}$ values over 3 to 5 events. The resulting

spread of the log($E_{dks}$) values (-1 to 1 for period A, -1 to 2 for period B) is roughly analogous to the spread of the log($E_d$) values (-0.5 to 0.5 for period A, -0.5 to 1 for period B; Rickenmann, 2020, Fig. 6 therein) for the event-based analysis, for which the log($E_d$) values were smoothened with a moving average over five events, before determining the cyclic variations (Rickenmann, 2020).

## 2.7 Hysteresis index

Several dimensionless numbers were developed and discussed to characterise the hysteresis between hydrological variables at the discharge event timescale (Lloyd et al., 2016; Zuecco et al., 2016). The index of Lloyd et al. (2016) was developed in the context of analyzing the hysteresis behaviour of suspended sediment measurements, and it was later applied also by Misset et al. (2018) and by Vale and Dymond (2019). It essentially calculates the difference of suspended sediment values on the rising and falling limbs and normalises the differences at every measurement point. This results in an index between -

1 and 1, that is equal to 0 if there is no loop.

I have tested some of the approaches for the bedload transport events at the Erlenbach, comparing the index values with a visual assessment of the hysteresis direction and magnitude. I found that the index proposed by Lloyd et al. (2016) works well when applied to bedload transport intensities using logarithmic (instead of linear) values of the impulse counts (*IMP*) to calculate differences and normalise these. Thus, the hysteresis index *HI_log* was calculated as summarised in Appendix A2.



## 3 Results

### 3.1 Characteristic ranges of discharges and disequilibrium ratios

The general trend of measured and calculated bedload transport rates in the Erlenbach is similar (Fig. 2). However, there is a wider spread of discharges for a given level of bedload transport in period B than in period A (Fig. 2), and for $Q < 0.5$ m$^3$ s$^{-1}$ the binned means of the $Q_{bM}$ values show a different trend for period B than for period A. A more detailed analysis of the somewhat different discharge and bedload transport observations for the two periods, particularly for $Q < 0.5$ m$^3$ s$^{-1}$, is presented in the in the Supporting information (Text S1, Fig. S2-S8). The main findings from this analysis are: (i) for $Q$ smaller than 1 m$^3$ s$^{-1}$, but particularly for $Q < 0.5$ m$^3$ s$^{-1}$, there were relatively more $Q_{bz} = 0$ values and relatively more small $Q_{bM}$ values in period A than in period B; (ii) in contrast, for $Q < 1$ m$^3$ s$^{-1}$, the average $Q$ values in period A were larger than those in period B; (iii) for $Q > 1$ m$^3$ s$^{-1}$, both $Q$ and $Q_{bM}$ values were smaller on average in period A than in period B.

To further compare the flow and transport conditions for period A and period B, binned values over 0.1 m$^3$ s$^{-1}$ wide discharge classes were determined for three variables, as shown in Fig. 3. The measured ($Q_{bM}$) and calculated ($Q_{btot}$) bedload transport rates are shown as time-integrated values, by multiplying the binned means by 60 s and summing them over the number of values per bin: $Q_{bM\_sum}$, $Q_{btot\_sum}$; and for the disequilibrium ratios the mean values were determined with the logarithmic values ($LE_{dM\_mean}$). In Fig. 3 we can observe three characteristic discharge ranges (S, M, H), and characteristic values for these three ranges are summarised in Table 1. In the domain S (small discharges, $Q < 0.5$ m$^3$ s$^{-1}$) there is a large deviation of the $E_{dM}$ values from the mean for both periods, which may be partly due to measurement uncertainties, whereas in the other domains the $E_{dM}$ values fluctuate around 1. The time-integrated transport in domain S ($Q_{bz\_sum}$ values in Table 1) is negligible in relation to the total for each period. The domain M (medium discharges, 0.5 m$^3$ s$^{-1}$ < $Q$ < 1.8 m$^3$ s$^{-1}$) is characterised by a similar bedload transport behaviour for periods A and B, with almost the same transported bedload masses in both periods, and similar binned mean of $E_{dM}$ values and hydraulic forcing ($Q_{btot\_sum}$ values in Table 1). In the domain H (high discharges, $Q > 1.8$ m$^3$ s$^{-1}$) we observe roughly similar flow and bedload transport characteristics for both periods for $Q$ up to about 5 m$^3$ s$^{-1}$ (Fig. 2, Fig. 3). However, in period B there were two exceptional flood events (Table 2) with $Q_{max} > 5$ m$^3$ s$^{-1}$, whereas in period A there was only one such exceptional flood event (Table 2), with only about half the transport time duration.

The roughly parallel lines of $Q_{bM\_sum}$ and $Q_{btot\_sum}$ in Fig. 3 indicate that there is a certain correlation between the measured bedload transport and the driving force discharge (expressed in terms of calculated transport). To assess how the correlation between flow and transport increases with integration time, the Pearson correlation coefficient $R$ was determined between $\log(Q_{bM})$ and $\log(Q_{btot})$ (Fig. 4). This calculation was done separately for periods A and B, but also for subsets including sub-periods with only above ("high", EDH) and below ("low", EDL) median $E_{dks}$ conditions, respectively. This analysis resulted in two important differences: Correlation is larger in both periods for EDH conditions than for EDL conditions, and correlation is larger in period A than in period B.



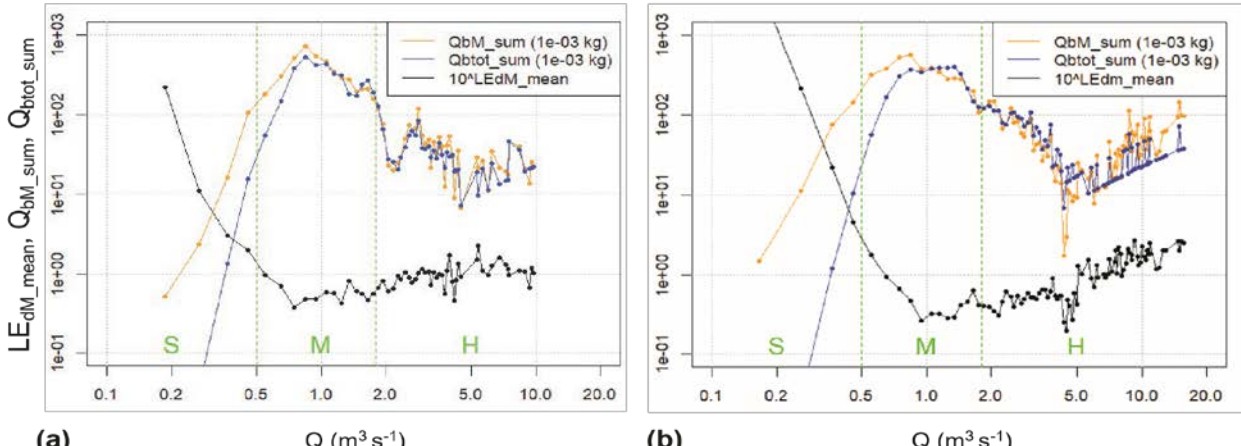

**Figure 3. Binned values over 0.1 m³ s⁻¹ wide discharge classes are shown for the measured ($Q_{bM}$) and calculated ($Q_{btot}$) bedload transport rates as time-integrated values ($Q_{bM\_sum}$ and $Q_{btot\_sum}$) and for the logarithmic means of the disequilibrium ratios ($LE_{dM\_mean}$), separately for (a) period A and (b) period B. The vertical, dashed green lines separate the three characteristic ranges (S, M, H) of discharge (see text for more details).**

**Table 1. Characteristic bedload transport values determined for the three characteristic discharge ranges (S, M, H), as delineated in Fig. 3.**

| Period | No. min values | Proportion of period | Qbtot_sum, (10⁶ kg) | Qbz_sum, (10⁶ kg) | EdM_ geo-mean | EdM_ median |
|---|---|---|---|---|---|---|
| **range (S):** | **Q < 0.5 m³ s⁻¹** | | | | | |
| A | 8832 | 0.23 | 0.0168 | 0.124 | 2.52 | 3.13 |
| B | 6399 | 0.19 | 0.0114 | 0.226 | 10.5 | 9.22 |
| ratio A/B | 1.38 | | 1.48 | 0.55 | 0.24 | 0.34 |
| **range (M):** | **0.5 m³ s⁻¹ < Q < 1.8 m³ s⁻¹** | | | | | |
| A | 28661 | 0.76 | 3.64 | 4.38 | 0.59 | 0.7 |
| B | 26633 | 0.79 | 3.60 | 4.27 | 0.71 | 0.77 |
| ratio A/B | 1.08 | | 1.01 | 1.03 | 0.83 | 0.91 |
| **range (H):** | **1.8 m³ s⁻¹ < Q** | | | | | |
| A | 419 | 0.01 | 1.35 | 1.55 | 0.83 | 0.86 |
| B | 795 | 0.02 | 3.08 | 3.7 | 0.49 | 0.56 |
| ratio A/B | 0.53 | | 0.43 | 0.42 | 1.71 | 1.52 |

The logarithmic values of the smoothened disequilibrium ratios ($E_{dks}$) are shown over time (no. of minute time steps) for

both periods in Fig. 5. In this figure a cyclic fluctuation of the $E_{dks}$ values can be observed, similarly to what had been found previously for the event-based analysis of the disequilibrium ratio for the same observation periods in the Erlenbach (Rickenmann, 2020). The cyclic fluctuations of $E_{dks}$ in Fig. 5 were used as a basis to delineate sub-periods (p1-p13) with primarily EDH or EDL conditions, and the limits were determined in such a way that they were identical with flood-event limits. This resulted in 7 sub-periods for period A and in 6 sub-periods for period B, respecting as a further criterion that all

these sub-periods should include a similar number (within a factor of about 2) of 1 min time steps (Table 2).



**Table 2. Identified sub-periods (see also Fig. 5) and their main characteristics in terms of number of minute values, number of flood events, mean $E_{dM}$ level, peak flow ($Q_{max}$) of the most important flood per sub-period, and the dates of the three exceptional flood events. The geometric mean of the 1 min EdM values (\*) is also given, as compared to the geometric mean of the seven (period A) and six (period B) mean values per sub-period (\*\*) given in this table.**

| Period, sub-period | Begin date of first event | End date of last event | No. min values | No. events | Geometric mean of EdM | Qmax (m³ s⁻¹) | Date exceptional flood event |
|---|---|---|---|---|---|---|---|
| **Period A** | | | | | | | |
| p1 | 20 October 1986 21:46 | 19 December 1987 09:09 | 5431 | 33 | 1.45 | 4.07 | |
| p2 | 10 May 1988 22:25 | 20 March 1991 07:08 | 6301 | 52 | 0.31 | 2.52 | |
| p3 | 03 June 1991 07:17 | 31 July 1993 07:22 | 5596 | 48 | 0.83 | 4.46 | |
| p4 | 10 August 1993 16:02 | 02 September 1994 04:34 | 3983 | 25 | 0.43 | 1.86 | |
| p5 | 08 September 1994 18:09 | 05 July 1996 18:07 | 6486 | 42 | 2.25 | 9.75 | 14 July 1995 |
| p6 | 08 July 1996 00:55 | 22 August 1998 08:01 | 5997 | 49 | 1.17 | 4.34 | |
| p7 | 22 August 1998 21:13 | 30 September 1999 14:15 | 4136 | 37 | 0.41 | 1.98 | |
| **Total / mean** | | | **37930** | **286** | *0.779 (\*\*)* | | |
| | | | | | *0.826 (\*)* | | |
| **Period B** | | | | | | | |
| p8 | 16 November 2002 11:20 | 17 September 2006 10:42 | 4443 | 30 | 0.15 | 3.76 | |
| p9 | 17 May 2007 20:12 | 14 July 2008 11:02 | 6002 | 34 | 7.99 | 15.61 | 20 June 2007 |
| p10 | 20 July 2008 13:53 | 29 July 2010 12:24 | 6699 | 45 | 1.35 | 4.99 | |
| p11 | 30 July 2010 06:34 | 10 August 2011 00:49 | 6437 | 41 | 2.09 | 10.89 | 01 August 2010 |
| p12 | 15 August 2011 02:45 | 29 August 2014 17:49 | 5947 | 47 | 0.35 | 4.79 | |
| p13 | 31 August 2014 17:53 | 26 October 2016 09:05 | 4389 | 39 | 1.18 | 2.00 | |
| **Total / mean** | | | **33917** | **236** | *1.058 (\*\*)* | | |
| | | | | | *1.170 (\*)* | | |

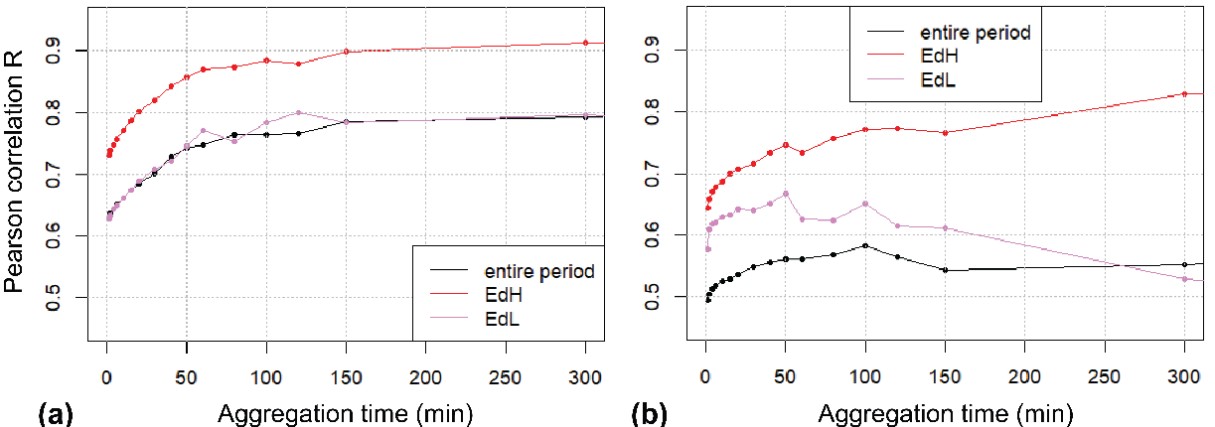

**Figure 4. Pearson correlation $R$ between $\log(Q_{bM})$ and $\log(Q_{btot})$ as a function of aggregation time, for (a) period A, and (b) period (B). EDH and EDL refer to sub-periods with above ("EDH, high") and below ("EDL, low") median $E_{dks}$ conditions, respectively.**





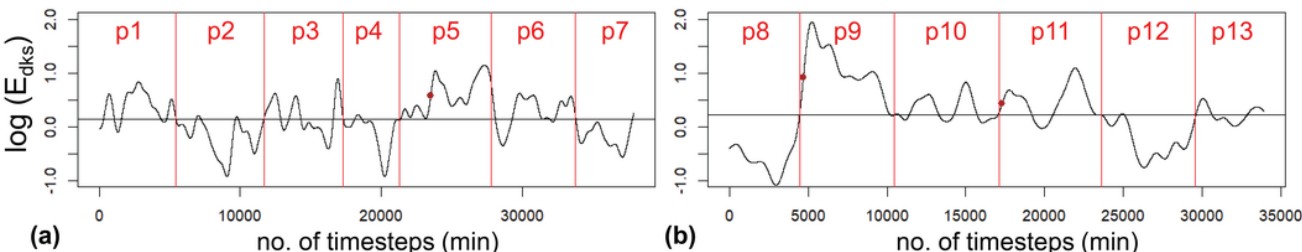

**Figure 5. Smoothened disequilibrium ratio ($E_{dks}$) as a function of the number (no.) of time steps, for (a) period A and (b) period B. The vertical red lines delineate sub-periods, for which the limits are identical with event limits. The red dots refer to the three exceptional flood events. The horizontal black line indicates the median $E_{dks}$ value for period A and B, respectively.**

The generally stronger correlation between $\log(Q_{bM})$ and $\log(Q_{btot})$ in period A than in period B (Fig. 4) may be partly due to a wider spread of discharges for a given level of bedload transport in period B than in period A (Fig. 2). The wider spread of discharges and the more horizontal distribution of binned $Q_{bM}$ values for $Q < 0.5$ m$^3$ s$^{-1}$ (Fig. 2) may reflect a wider range of "phase 1" transport conditions (whereby transport rates are relatively low until a certain flow level is reached, Ryan et al., 2002) for period B than for period A, as indicated by the smoothened trend lines for $Q_{bM}$ versus $Q$ that were determined

separately for each sub-period of Table 2 (Fig. S9). The observation of a stronger correlation between $\log(Q_{bM})$ and $\log(Q_{btot})$ for EDH conditions than for EDL conditions (Fig. 4) may be partly due to the fact that there were relatively fewer $Q_{bz} = 0$ values for EDH conditions than for EDL conditions, as is indicated in Table 3.

**Table 3. Separation of all bedload transport observations into two subsets of equal number of observations per (main) period, with either above ("high", EDH) or below ("low", EDL) median $E_{dM}$ conditions. For these subsets, the table indicates the proportion of values with $Q_{bz} = 0$, i.e. the percentage of values when zero bedload transport was measured during 1 min intervals of the transport events.**

| Label | Condition | Proportion of values with Qbz=0 | |
| --- | --- | --- | --- |
| | | Period A | Period B |
| "EdH" | EdM > median(EdM) | 16.0% | 8.3% |
| "EdL" | EdM < median(EdM) | 20.0% | 13.1% |

**3.2 Autocorrelation coefficient of bedload transport rates as a function of lag time**

Given a median flood event duration of about 95 min to 100 min for both periods (Fig. S1), I determined the autocorrelation coefficient ($ACF$) of $\log(Q_{bM})$ and of $\log(E_{dM})$ for lag times up to 100 min, separately for periods A and B (Fig. 6). The main findings of this analysis are: The $ACF$ values of $\log(Q_{bM})$ are roughly similar for both periods for lag times up to about 50 min (Fig. 6 a, b), and in period B the two sub-periods with the two exceptional flood events (p9, p11) have among the largest

$ACF$ values. Regarding the $ACF$ values of $\log(E_{dM})$, they are clearly larger for period B than for period A (Fig. 6 c, d), and in





both main periods the sub-periods with the largest exceptional flood events (p5, p9, p11) have among the largest *ACF* values for lag times up to 100 min.

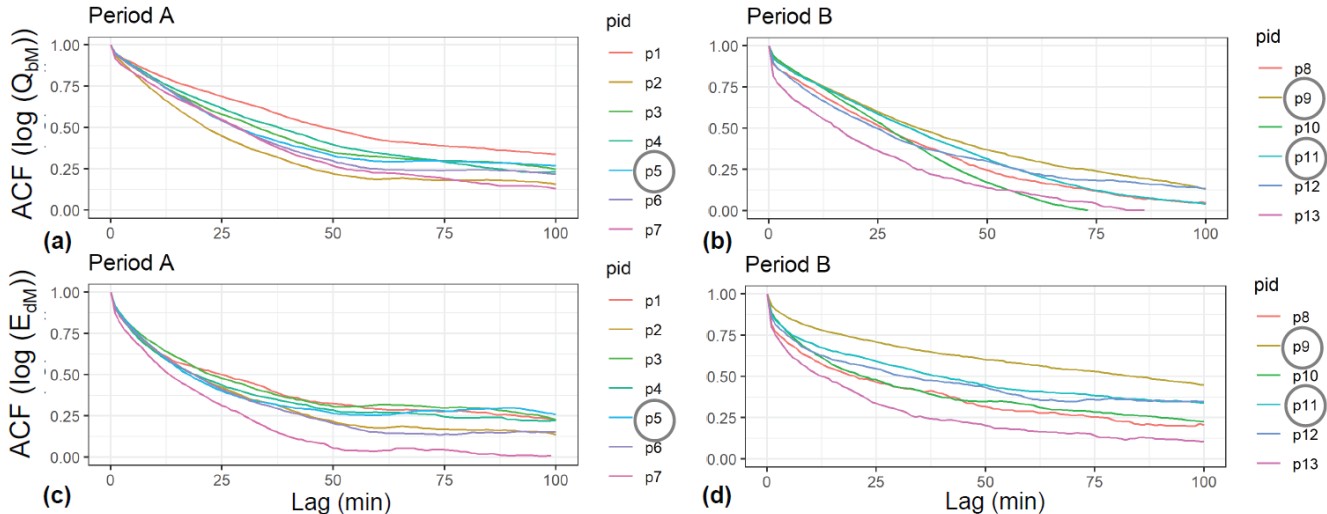

**Figure 6. Autocorrelation coefficient (*ACF*) of log($Q_{bM}$) (a, b) and *ACF* of log($E_{dM}$) (c, d) for the main sub-periods indicated in Fig. 5. "pid" refers to the identification of the sub-period. The gray circles indicate inclusion of one of the three exceptional flood events.**

A correlation matrix was determined for seven variables for the 13 sub-periods in Table 2 (Fig. 7a), namely: the mean *ACF* values over lag times up to 30 min for the log values of $Q_{btot}$, $Q_{bM}$, and $E_{dM}$ (ACF_LQbt_m30, ACF_LQbM_m30, ACF_LEdM_m30); the mean of the log values of $Q_{btot}$, $Q_{bM}$, and $E_{dM}$ (LQ$_{btot\_mean}$, LQ$_{bM\_mean}$, LE$_{dM\_mean}$); and the mean of the linear values of $Q_{bM}$ (Q$_{bM\_mean}$). The strongest correlation for the mean *ACF* values was found between ACF_LE$_{dM\_m30}$ and Q$_{bM\_mean}$, which is graphically illustrated in Fig. 7b. There the two sub-periods with the two largest exceptional flood events

(p9, p11) contribute much to the strong correlation, although the data from the other sub-periods support the general trend line.

The fields with a yellow box in the correlation matrix of Fig. 7a indicate those variable combinations with a reasonably strong correlation, for which the correlation has the same sign (positive or negative) also when considering only period A and period B separately (Fig. S10). However, the correlation between the log values of $E_{dM}$ (ACF_LE$_{dM\_m30}$) and the log

values of $Q_{bM}$ (LQ$_{bM\_mean}$) is clearly weaker than for the case of using linear values of $Q_{bM}$ (Q$_{bM\_mean}$) (Fig. 7a). In addition the sign of the correlation between ACF_LQ$_{bt\_m30}$ and LQ$_{bM\_mean}$ is different for period A and B (Fig. S10). This is an indication that only larger transport events (that dominate the linear averaging) clearly increase autocorrelation and thus memory effects.




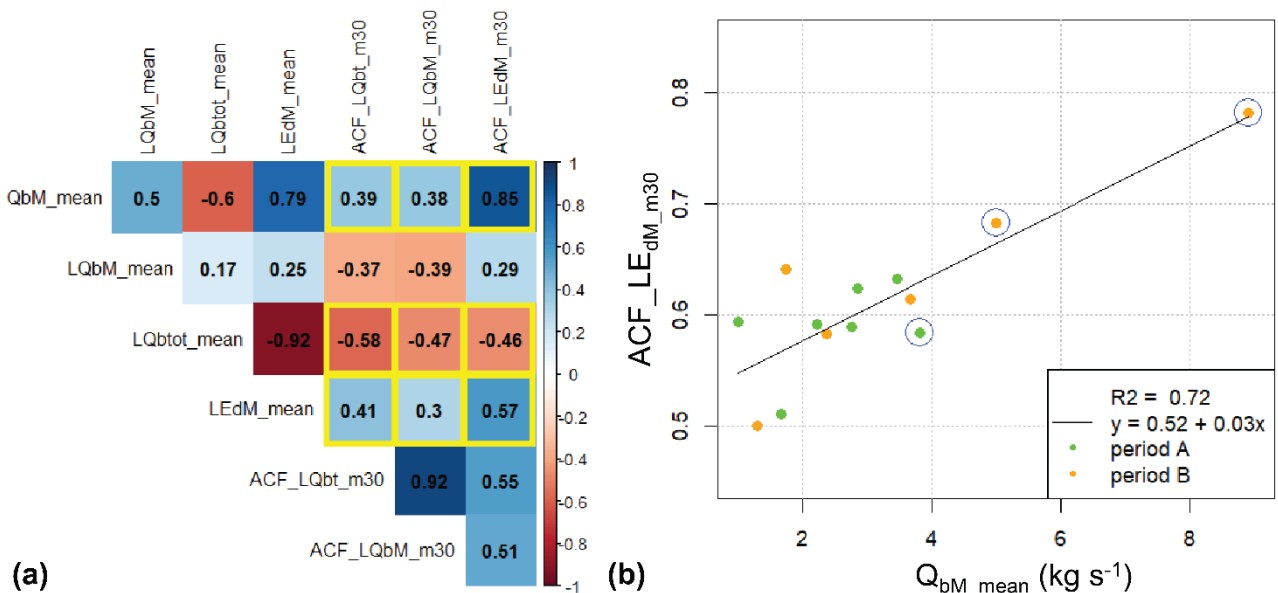

**Figure 7. (a) Correlation matrix with Pearson correlation coefficient *R*, for seven variables determined for all the 13 sub-periods indicated in Fig. 5 and Table 2. The fields with a yellow box indicate those variable combinations with a reasonably strong correlation, for which the correlation has the same sign (positive or negative) also when considering only period A and period B separately (see Fig. S10). (b) Mean autocorrelation coefficient of log($E_{dM}$) over the first 30 lag minutes (ACF_LE$_{dM\_m30}$) vs. linear mean of Q$_{bM}$ values (Q$_{bM\_mean}$), determined for each of the 13 sub-periods. The blue circles around a point indicate the inclusion of one of the three exceptional flood events.**

### 3.3 Critical discharge at the begin and at the end of a transport event

For the 13 subperiods (Table 2), the means of both the critical discharge at the begin and at the end of a transport event show a fairly strong correlation with the geometric means of the $E_{dM}$ values (Fig. 8). For comparison, it is noted that for the event-based analysis in Rickenmann (2020) the squared correlation coefficient between $Q_e$ and $E_d$ is $R^2 = 0.26$, and between $Q_s$ and $E_d$ it is $R^2 = 0.20$. These $R^2$ values are clearly lower than those given in Fig. 8. The analysis based on the 13 sub-periods represents on average an aggregation time 40-times longer than for the event-based analysis (the factor 40 is determined as the ratio of 522 events over 13 sub-periods). Similarly, there is a fairly strong correlation between the means of both the critical discharge at the begin and at the end of a transport event and the geometric means of $Q_{btot}$ (Fig. 9).

If we consider the correlation between $Q_e$ and $Q_{bM}$ in Fig. 10a, it is clearly lower than the correlations of $Q_e$ with log(EdM) and log($Q_{btot}$) in Fig. 8 and Fig. 9, respectively. Calculating the mean values per sub-period with the geometric instead of the




linear mean of the $Q_{bM}$ values, results in a vanishing correlation between the two variables (Fig. 10b). These findings are consistent with the event-based analysis in Rickenmann (2020).

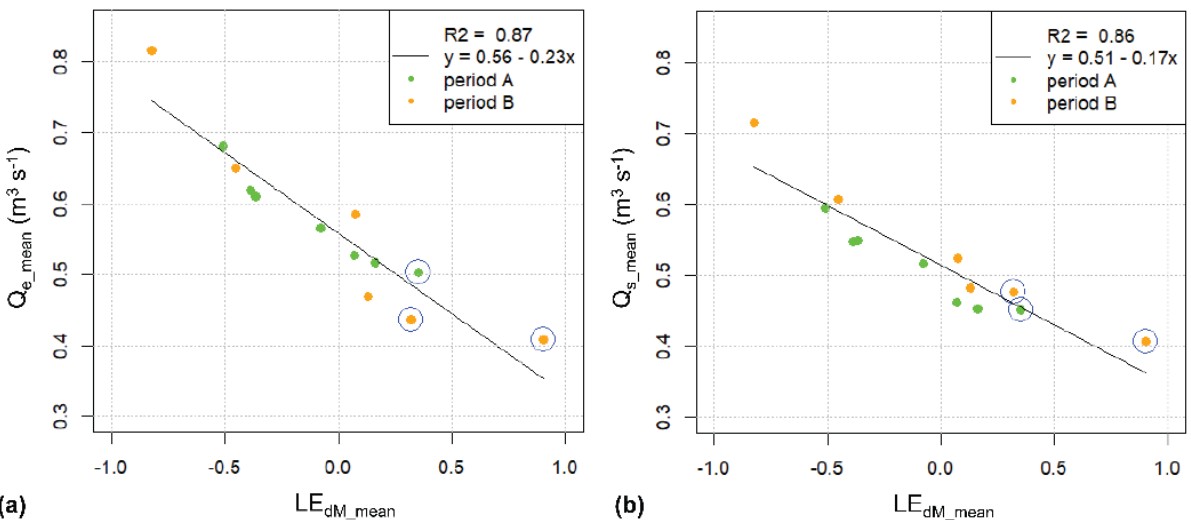

**Figure 8. Correlation between variables determined for each of the 13 sub-periods. (a) Mean discharge threshold at the end of an event $Q_e$ ($Q_{e\_mean}$) vs. the geometric mean of the disequilibrium ratio $E_{dM}$ ($LE_{dM\_mean}$), (b) Mean discharge threshold at the start of an event $Q_s$ ($Q_{s\_mean}$) vs. the geometric mean of the disequilibrium ratio $E_{dM}$ ($LE_{dM\_mean}$). The blue circles around a point indicate inclusion of one of the three exceptional flood events.**

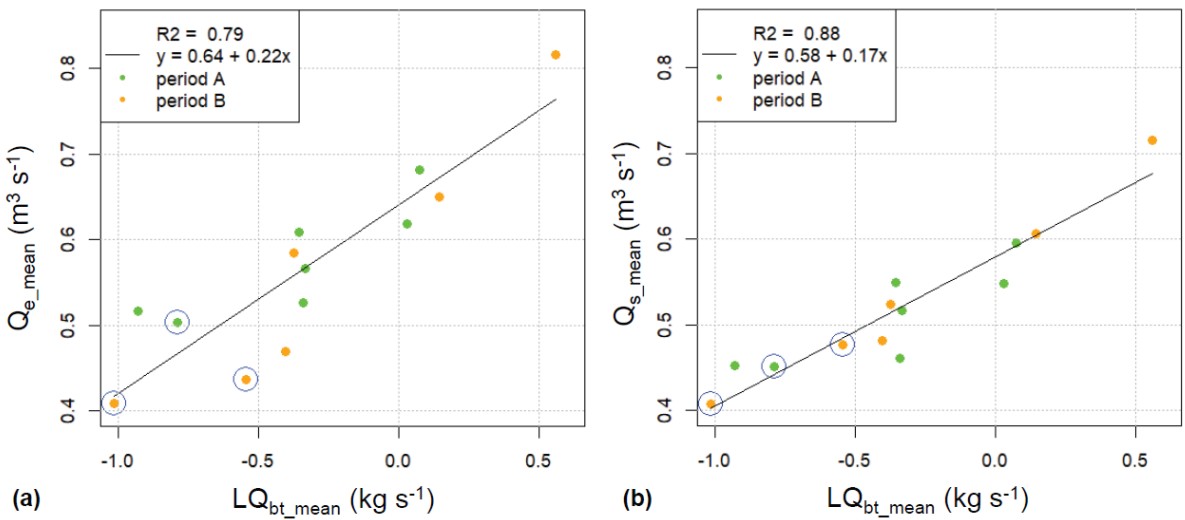

**Figure 9. Correlation between variables determined for each of the 13 sub-periods. (a) Mean discharge threshold at the end of an event $Q_e$ ($Q_{e\_mean}$) vs. the geometric mean of the calculated transport rate $Q_{btot}$ ($LQ_{bt\_mean}$), (b) Mean discharge threshold at the start of an event $Q_s$ ($Q_{s\_mean}$) vs. the geometric mean of the calculated transport rate $Q_{btot}$ ($LQ_{bt\_mean}$). The blue circles around a point indicate inclusion of one of the three exceptional flood events.**





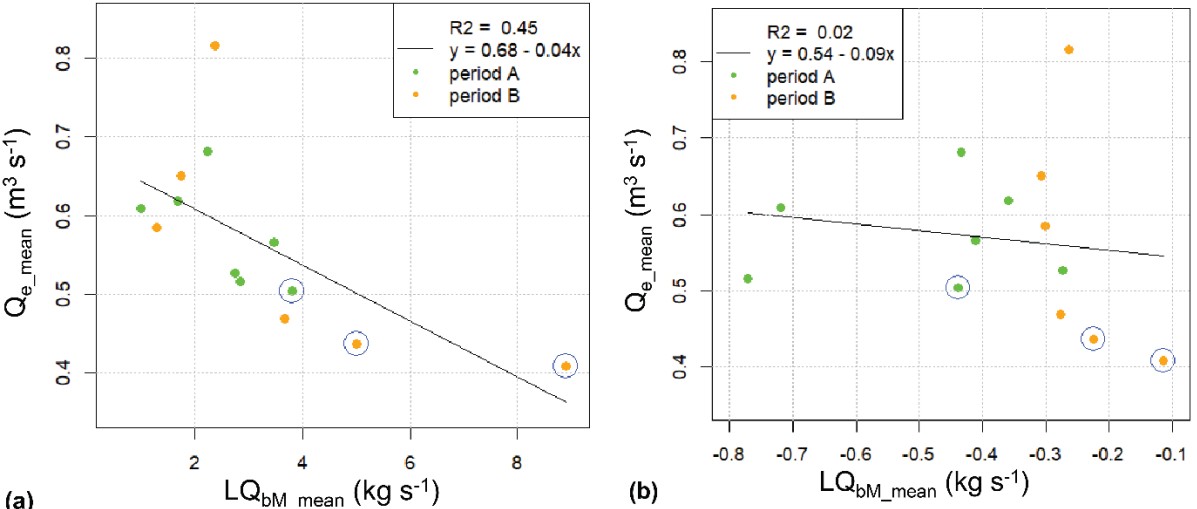

**(a)**

**(b)**

**Figure 10. Correlation between variables determined for each of the 13 sub-periods. (a) Mean discharge threshold at the end of an event $Q_e$ ($Q_{e\_mean}$) vs. the linear mean of the bedload transport rate $Q_{bM}$ ($Q_{bM\_mean}$), (b) Mean discharge threshold at the end of an event $Q_e$ ($Q_{e\_mean}$) vs. the geometric mean of the bedload transport rate $Q_{bM}$ ($LQ_{bM\_mean}$). The blue circles around a point indicate inclusion of one of the three exceptional flood events.**


### 3.4 Coefficient of variation of the bedload transport rates

Since the discharge $Q$ (or hydraulic forcing) is the primary control on bedload transport rates, the coefficient of variation ($cv$) of the bedload transport rates $Q_{bM}$ ($cv\_Q_{bM}$) was determined for $Q$-ordered values. Each bin had a constant width of approximately 200 values, and both $cv\_Q_{bM}$ and the linear mean of the bedload transport rates $Q_{bM}$ ($Q_{bM\_mean}$) were

calculated. The $cv$ values were determined for the following two cases: (i) Considering period A and period B separately, and (ii) for both periods A and B combined, the data was divided into one group pertaining to the rising limb and a second pertaining to the falling limb of the hydrograph (denoted as HR and HF, respectively) of each flood event, taking the maximum discharge of an event to separate the two limbs. (A theoretically more correct calculation of $cv$ values was made also with $Q_{bz}$ instead of $Q_{bM}$ values. This resulted in very slightly increased $cv$ values, mainly for period A at small

discharges, with more $Q_{bz} = 0$ values in period A than in period B. For the sake of simplicity, the $cv\_Q_{bM}$ values are presented here, since all the other analyses were made and presented with $Q_{bM}$ values.)

For case (i) $cv\_Q_{bM}$ values vary essentially between about 1 and 3. In terms of discharge, three ranges can be distinguished. For both 0.4 m$^3$ s$^{-1}$ < $Q_{\_mean}$ < 0.6 m$^3$ s$^{-1}$ and for 0.9 m$^3$ s$^{-1}$ < $Q_{\_mean}$ < 1.6 m$^3$ s$^{-1}$ the cv values are significantly smaller for period A than for period B (Fig. 11a, 11c), whereas cv values are not significantly different between the two periods for an

intermediate discharge range 0.6 m$^3$ s$^{-1}$ < $Q_{\_mean}$ < 0.9 m$^3$ s$^{-1}$ (Fig. 11b). For the lowest discharge range, the larger $cv$ values for period B are associated with significantly larger bedload transport rates in period B than in period A (Fig. 12a). In the other (higher) two discharge ranges there is no significant difference in the bedload transport rates (Fig. 12b, 12c).





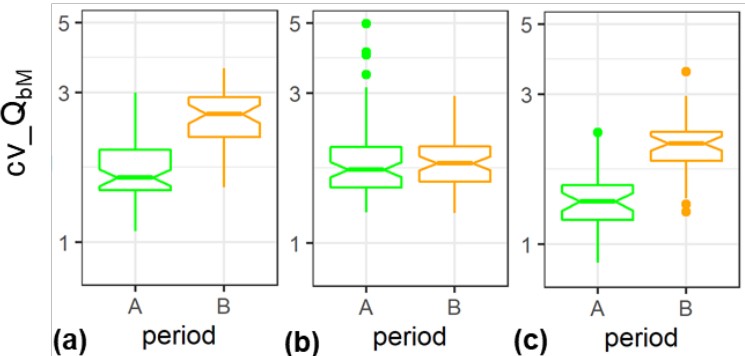

**Figure 11. Boxplots of the coefficient of variation of bedload transport rates ($cv\_Q_{bM}$) determined for binned $Q$ values, each containing 200 values, shown for different discharge ranges: (a) 0.4 m$^3$ s$^{-1}$ < $Q_{\_mean}$ < 0.6 m$^3$ s$^{-1}$, (b) 0.6 m$^3$ s$^{-1}$ < $Q_{\_mean}$ < 0.9 m$^3$ s$^{-1}$, (c) 0.9 m$^3$ s$^{-1}$ < $Q_{\_mean}$ < 6 m$^3$ s$^{-1}$.**

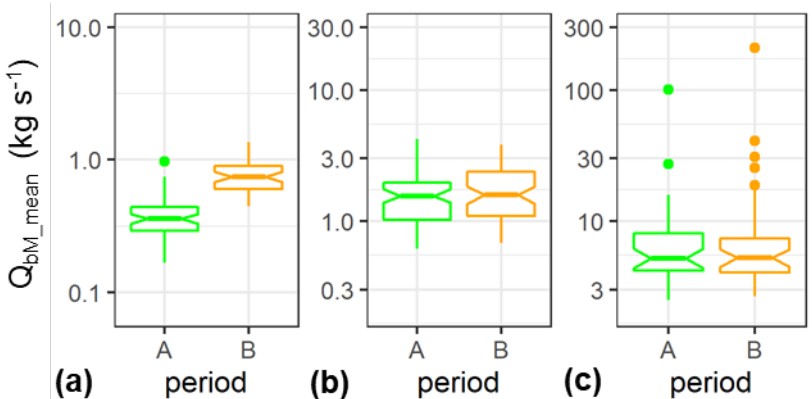

**Figure 12. Boxplots of the linear mean of bedload transport rate ($Q_{bM\_mean}$) determined for binned $Q$ values, each containing 200 values, shown for different discharge ranges: (a) 0.4 m$^3$ s$^{-1}$ < $Q_{\_mean}$ < 0.6 m$^3$ s$^{-1}$, (b) 0.6 m$^3$ s$^{-1}$ < $Q_{\_mean}$ < 0.9 m$^3$ s$^{-1}$, (c) 0.9 m$^3$ s$^{-1}$ < $Q_{\_mean}$ < 6 m$^3$ s$^{-1}$. Note different ordinate values, but similar scaling to allow an easy visual comparison of the spread of the values in the different discharge ranges.**


For case (ii) using the groups of HR and HF values, $cv\_Q_{bM}$ values are significantly smaller on average for the rising limb (HR) than for the falling limb (HF) flow conditions, whereby the relative difference is more pronounced for discharges $Q_{\_mean}$ > 1 m$^3$ s$^{-1}$ than for $Q_{\_mean}$ < 1 m$^3$ s$^{-1}$ (Fig. 13a,13b). For increasing discharges larger than about 0.5 m$^3$ s$^{-1}$, the $cv$ values tend to decrease (Fig. S11). Furthermore, for a discharge range of about 0.5 m$^3$ s$^{-1}$ < $Q_{\_mean}$ < 1.8 m$^3$ s$^{-1}$, a predominant

clockwise hysteresis behaviour appears to be most pronounced, as reflected by generally larger average $Q_{bM}$ values for the rising than for the falling limb (Fig. 13c).



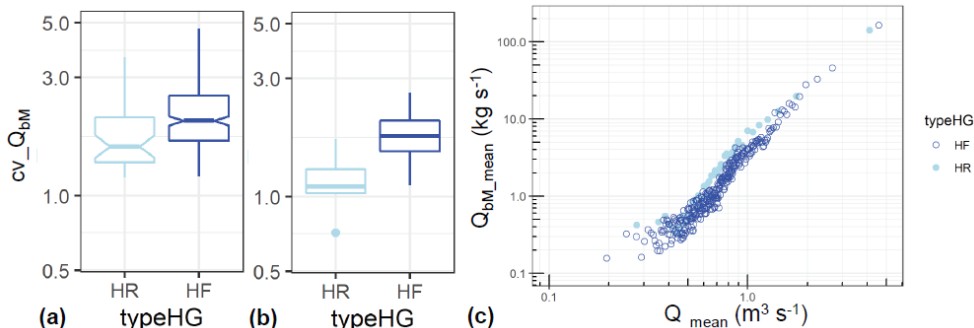

**Figure 13. Boxplots of the coefficient of variation of bedload transport rates ($cv\_Q_{bM}$) for the rising part (HR) and the falling part (HF) of the hydrograph, shown (a) for smaller discharges $Q\_mean < 1$ m³ s⁻¹, and (b) for larger discharges $Q\_mean > 1$ m³ s⁻¹. (c) Linear mean of bedload transport rates ($Q_{bM\_mean}$) vs. linear mean of discharge ($Q\_mean$), determined separately for HR and HF conditions.**

## 3.5 Hysteresis index

The hysteresis index *HI*_log generally varies between -0.5 and 0.75 for both periods A and B (Fig. 14). For increasing $Q_{max}$ values, a decrease in the spread of the *HI*_log values is observed in the range of discharges of about 1 m³ s⁻¹ $< Q_{max} < 2$ m³ s⁻¹ (Fig. 14a). A similar decrease in the spread of the *HI*_log values occurs in the range of event bedload masses ($M_{gravel}$) of about 1 10⁴ kg $< M_{gravel} < 1$ 10⁵ kg (Fig. 14b). It is further observed that the *HI*_log values for period B are significantly smaller on average than for period A (Fig. S12), with the *HI*_log values changing somewhat for different discharge ranges

(S, M, H; see Fig. 3). In the high discharge range ($Q > 1.8$ m³ s⁻¹), the values indicate less clockwise behaviour and the difference between period A and B is largest (Fig. S12c). The hysteresis analysis clearly shows a dominant clockwise hysteresis pattern in the Erlenbach stream, and the largest *HI*_log values are generally observed in the intermediate discharge range 0.5 m³ s⁻¹ $< Q\_mean < 1.8$ m³ s⁻¹ (Fig. 14, Fig. 13c, Fig. S12).

The lower limit for the start of the decreasing spread of *HI*_log values (Fig. 14b), $M_{gravel} = 1$ 10⁴ kg, corresponds to a bulk

bedload volume (including porosity) of about 7.4 m³. For the lowermost natural reach with $b_w = 4.1$ m and a length of about 50 m, this roughly corresponds to a mean thickness $h_m$ of a (mobile) bedload layer stored in this reach of about $h_m = 7.4$ m³/205 m² = 0.04 m. The upper limit of the decreasing spread of *HI*_log values (Fig. 14b), $M_{gravel} = 1$ 10⁵ kg, is equivalent to a bulk bedload volume (including porosity) of about 74 m³, corresponding to a mean thickness $h_m$ of about 0.36 m. These two values for $h_m$ are of the order of the $D_{50}$ and the $D_{84}$ of the surface bed material, respectively.

A possible effect of channel destabilization on the hysteresis behaviour is shown in Fig. 14c. The normalised maximum discharge of the preceding flood event ($Q_{max\_p}$)/$Q_{max}$ is shown to influence the degree of hysteresis (*HI*_log value), in that for values of about ($Q_{max\_p}$)/$Q_{max} > 1.5$ and $Q_{max} > 1.3$ m³ s⁻¹, there are very few events with a strong clockwise hysteresis (*HI*_log $> 0.4$, i.e. *HI*_log $+0.5 > 0.9$). This implies that antecedent large magnitude floods tended to result in less clockwise hysteresis in the following flood event with a minimum size of $Q_{max} > 1.3$ m³ s⁻¹.


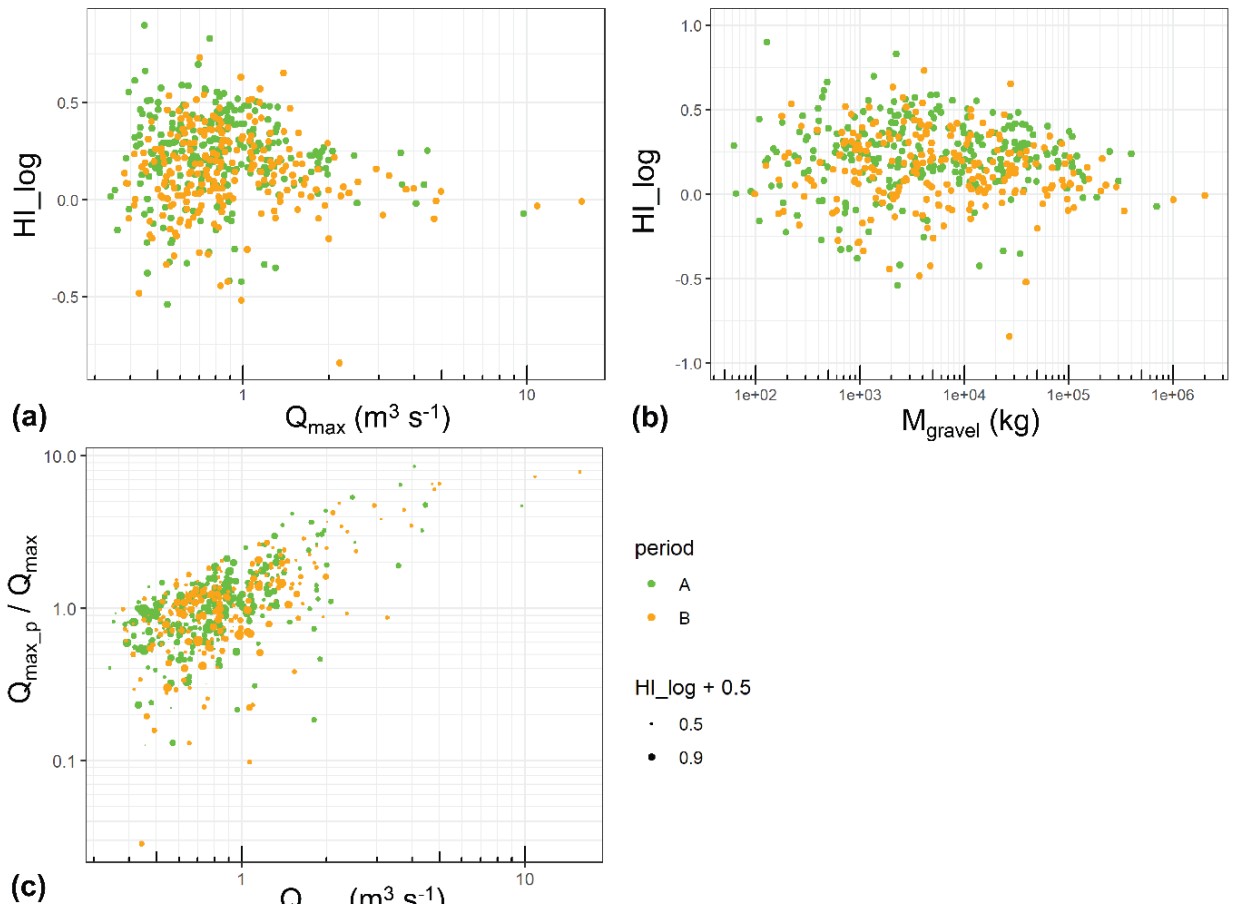

**Figure 14. (a) Hysteresis index *HI*_log vs. maximum discharge *Q*$_{max}$ of an event. (b) Hysteresis index *HI*_log vs. total transported bedload mass of an event (*M*$_{gravel}$). (c) Effect of normalised maximum discharge of preceding flood event (Q$_{max\_p}$)/*Q*$_{max}$ shown vs. *Q*$_{max}$, with the degree of hysteresis (*HI*_log) indicated by the data point size; larger circle sizes indicate a stronger clockwise hysteresis.**

# 4 Discussion

## 4.1 Averaging and characteristic trends

In some parts of the analysis, different trends were found for transport conditions that imply either above average disequilibrium ratios or below average disequilibrium ratios. These conditions are identified as time steps with above 430 ("high", EDH) and below ("low", EDL) median $E_{dks}$ values, respectively. The event-based analysis of the Erlenbach sediment transport data had indicated that EDH-conditions are likely associated with a larger sediment availability on the streambed than EDL-conditions; in addition, clearly larger disequilibrium ratios were observed in period B than in period A,



associated with a larger volume of sediment stored on the streambed (Rickenmann, 2020). In previous studies, the
correlation between hydraulic forcing and bedload transport rate has been shown to increase with increasing aggregation

times (e.g., Lenzi et al., 2004; Recking et al., 2012; Rickenmann, 1994, 2018; Rickenmann and McArdell, 2008). This is
confirmed for the Erlenbach when considering maximum aggregation times of 300 minutes (with most flood events not
exceeding 300 minutes in duration, Fig. S1), for which the correlation coefficients $R$ also increase with increasing
aggregation times (Fig. 4). The analysis of the Erlenbach data further shows that the correlation between the two variables is
larger for above average bedload transport (EDH) than for below average bedload transport (EDL) (Fig. 4). This statement

appears to contradict the fact that $R$ values are generally smaller for period B than for period A (Fig. 4), because larger
volumes of sediment stored on the streambed were observed in period B than in period A (Rickenmann, 2020). However, the
smaller $R$ values for period B than for period A may be due to the more frequent occurrence of widely varying bedload
transport rates for small discharges during period B (Fig. 2). This is associated with more pronounced phase 1 transport
conditions in period B than in period A (Fig. S9), including larger fluctuations in $Q$ and $Q_b$ without a clear correlation

between the two variables.

In the event-based analysis of Erlenbach bedload transport measurements, the threshold discharge for transport ($Q_s$, $Q_e$) was
found to correlate positively with hydraulic forcing ($Q_{btot}$) and negatively with the disequilibrium ratio ($E_{dM}$) (Rickenmann,
2020). This finding is confirmed by the present analysis for the 13 sub-periods (Table 2, Fig. 5), for which the aggregation
times were about 40 times longer than for the event-based analysis, resulting in correlation coefficients $R^2$ between $Q_s$ or $Q_e$

and $Q_{btot}$ or $E_{dM}$ ranging from 0.88 to 0.79 (Fig. 8 and Fig.9), compared to correlation coefficients $R^2$ of up to 0.26 for the
event-based analysis (Rickenmann, 2020). If we consider the correlation between $Q_s$ or $Q_e$ and transport rate $Q_{bM}$, the $R$
values for a (negative) correlation were smaller than between $Q_s$ or $Q_e$ and $Q_{btot}$ or $E_{dM}$: For the event-based analysis a
correlation was found only for those events with a peak discharge $Q$max larger than 1 m$^3$ s$^{-1}$ (Rickenmann, 2020), and for the
new analysis a correlation was found only when determining linear mean values of $Q_{bM}$ for each sub-period (Fig. 10). This

finding suggests that only flow conditions with more important bedload transport activity can influence the threshold
transport conditions.

In contrast to the event-based analysis (Rickenmann, 2020), the sub-periods defined in Fig. 5 may suggest longer full-cycle
durations. For period A, one could infer two full cycles (taking sub-periods p5 and p6 together), resulting in cycle duration
of about 6.5 years (as compared to 0.9 years for the the event-based analysis). For period B, one could infer one very long

cycle of about 10 years (sub-periods p8 to p11) and one of 4 years (as compared to 1.6 years for the the event-based
analysis). However, the delineation of the sub-periods in this study was made to explore the effect of an averaging over
aggregation periods longer than an event, and not to examine the cycle durations.

The autocorrelation coefficients $ACF$ for the 13 sub-periods for lag times up to about 30 to 50 minutes are generally
comparable for log($Q_{bM}$) for both periods A and B and for log($E_{dM}$) for period B, whereas these ACF values a somewhat

smaller for log($E_{dM}$) for period A (Fig. 6). For the examined $ACF$ values, the strongest correlation is found between $ACF$
values averaged over lag times of up to 30 minutes for the disequilibrium ratio (ACF_LE$_{dM\_m30}$) and the linear means of $Q_{bM}$



for each sub-period (Fig. 7b). Again, the correlation is clearly stronger when using the linear mean of $Q_{bM}$ values instead of the geometric (log) mean of $Q_{bM}$ values (Fig. 7a). This part of the analysis indicates that the memory effect for the disequilibrium ratio increases for periods with greater bedload transport activity, a finding in agreement with the event-based

analysis of the Erlenbach sediment transport data (Rickenmann, 2020) and with a flume study by Elgueta Astaburuaga et al. (2018). Autocorrelation values of $\log(E_{dM})$ are generally larger for periods with increased sediment availability (period B compared to A, and for sub-periods including extreme events, Fig. 6). These observations are in line with the positive correlation between ACF_LE$_{dM\_m30}$ and the linear means of Q$_{bM}$ values (Fig. 7b).

## 4.2 The role of sediment availability, hydraulic forcing, and bedload transport intensity on bedload transport
fluctuations

For discharges $Q$ smaller than about 0.6 m$^3$ s$^{-1}$ the $cv\_Q_{bM}$ values were smaller in period A than in period B (Fig. 11a), and the mean bedload transport rates $Q_{bM\_mean}$ were also smaller in period A than in period B (Fig. 12a). This is likely due to a higher sediment availability on the streambed in period B than in period A (Rickenmann, 2020), with more frequent phase 1 transport conditions in period B (Fig. S9), and with a more frequent occurrence of zero transport values in period A than in

period B (Table 3). However, the (higher) sediment availability appears to be insufficient for the transport capacity for phase 1 conditions, resulting both in larger $Q_{bM}$ values (more frequently in period B) and in smaller $Q_{bM}$ values (more frequently in period A, with more zero values). Phase 1 transport conditions were observed in several studies of mountain streams (Jackson and Beschta, 1982; Warburton, 1992; Ryan et al., 2002; Bathurst, 2007; Rickenmann, 2018).

In an intermediate range of discharges, roughly for 0.6 m$^3$ s$^{-1}$ < $Q_{mean}$ < 0.9 m$^3$ s$^{-1}$, which corresponds to about 0.25 kg s$^{-1}$ <
$Q_{bM}$ < 2.5 kg s$^{-1}$ (Fig. 2, Eq. A7), there is no clear difference in both the $cv\_Q_{bM}$ values and the $Q_{bM\_mean}$ values between periods A and B (Fig. 11b, Fig. 12b). This suggests that the generally higher sediment availability on the streambed in period B than in period A did not substantially affect the bedload transport fluctuations in this discharge range.

For discharges larger than 0.9 m$^3$ s$^{-1}$, the $cv\_Q_{bM}$ values were smaller in period A than in period B (Fig. 11c), whereas the mean bedload transport rates $Q_{bM\_mean}$ were similar in period A and in period B (Fig. 12c). It is hypothesized that for these

discharge conditions, the higher sediment availability on the streambed in period B than in period A (Rickenmann, 2020) resulted in more transport fluctuations, assuming that more larger, movable particles were available on the streambed in period B but their number on the bed was still too limited to result in a more continuous (approximately regular) high intensity transport.

The $cv$ values determined for the Erlenbach are partly similar to those derived from the flume and field studies cited in the
Introduction of this paper, but also include relatively frequent values of up to about 3. For the Erlenbach, the $cv$ values averaged over both periods A and B tended to decrease with increasing bedload transport rates only for larger flow intensities in period A (Fig. 11c). This is qualitatively consistent with trends observed in flume studies by Kuhnle and Southard (1988) and Mettra (2014) and in a field study by Kuhnle and Willis (1998).



At the Erlenbach, smaller $cv\_Q_{bM}$ values were observed for the rising limb than for the falling limb of the hydrograph (Fig.
13a, 13b). The mean relative duration of the rising limb is 12.7% of the total duration of a transporting flood event.
Therefore, it is hypothesized that during this relatively short time span the available sediment on the streambed was unlikely
to be quickly exhausted, in contrast to the much longer relative duration of the falling limb of 87.3% of the total hydrograph
duration. As a result, smaller $cv\_Q_{bM}$ values can be expected for the HR than for the HF conditions.

**4.3 Critical discharge and sediment availability to change transport conditions**

For larger discharges and more bedload transport, the strength of the (dominant) clockwise hysteresis in the Erlenbach was
reduced (Fig. 14, Fig. S12). For period B and for $Q > 1.8$ m$^3$ s$^{-1}$ the median hysteresis index $HI\_log$ was close to zero (i.e., no
hysteresis effect) and the spread of the $HI\_log$ values was reduced. According to Fig. 14b and section 3.4 there is a threshold
(transported) bedload volume of about 7.4 m$^3$ to 74 m$^3$, above which (clockwise) hysteresis effects tended to decrease. For
the lowermost 50 m long natural reach, these volumes correspond to a mean thickness of about 0.04 m to 0.36 m of stored
sediment on the bed, which is of the order of the estimated thickness of the mobile bedload layer in period A ($D_{50}$ to $D_{84}$,
Rickenmann, 2020). This is taken as an indirect support for the hypothesis that, given a sufficient sediment availability on
the streambed for a given hydraulic forcing ($Q > 1.8$ m$^3$ s$^{-1}$) to move the grains in this size range, the bedload transport
hysteresis in the Erlenbach should tend to disappear, assuming that higher flow intensity events increase sediment
availability by bank erosion and beginning step destruction (Golly et al., 2017). A partial confirmation of this idea is also
based on bedload transport measurements with a moving basket downstream of the 50 m reach, which indicate that for
discharges of up to 1.1 m$^3$ s$^{-1}$ no particles were transported with $D > 0.2$ m.

Clockwise hysteresis has been associated with a gradual decrease in sediment availability or early exhaustion of sediment
sources (Pretzlav et al., 2020; Mao et.al., 2019; Rovira and Batalla 2006; Gao and Pasternack, 2007). This explanation is in
qualitative agreement with our observations at the Erlenbach, where less clockwise hysteresis was observed for period B
than for period A (Fig. 14, Fig. S12), and where sediment availability was found to be larger in period B than in period A
(Rickenmann, 2020). That lower sediment availability results in a more pronounced clockwise hysteresis, is indirectly
confirmed in the Erlenbach by a (although weak) trend that increasing $HI\_log$ values tended to associated with increasing $Q_e$
values (Fig. S13), which were found to reflect a limited sediment availability on the bed. This finding agrees with flume
experiments of Mettra (2014), according to which a lower sediment supply, higher channel slope and lower Shields stress
lead to more intense hysteretic effects. Furthermore, from a review of hysteresis behaviour of sediment transport rates in
alluvial streams, Gunsolus and Binns (2018) concluded that lower magnitude hydrographs result in more pronounced
hysteresis of sediment transport rates than larger magnitude hydrographs. This finding also agrees with the observations at
the Erlenbach (Fig. 14, Fig. S12).

Previous studies for the Erlenbach indicate that bed strengthening by low flows or low to medium intensity sediment
transporting flows may be important for discharges up to about 1.4 m$^3$ s$^{-1}$ (Masteller et al., 2019; Rickenmann, 2020). This
tendency is supported here by the increasing threshold discharges at the start and end of an event for increasing discharge



intensities (Fig. 9). For larger discharges and for exceptional floods in the Erlenbach the mobility of streambed material appears to increase, as indicated by the decreasing threshold discharges for increasing bedload transport levels, and particularly including the exceptional flood events (Fig. 10). This latter observation is also supported by previous studies for

the Erlenbach: step migration was observed for discharges of 4 to 5 $m^3$ $s^{-1}$ Golly et al. (2017), and substantial destruction of steps was estimated to occur at a discharge of about 7 to 10 $m^3$ $s^{-1}$ (Turowski et al., 2009; Turowski et al., 2013). In the Rio Cordon torrent, a similar step-pool stream in Northern Italy, bedload transport rates had increased by about a factor of 5 following an exceptional flood event with a peak discharge of about 10 $m^3$ $s^{-1}$ (Lenzi et al., 2004; Rainato et al., 2017). In the Erlenbach, bedload transport rates had increased by roughly a factor of 10 following the exceptional flood event in June

2007 with a peak discharge of about 16 $m^3$ $s^{-1}$ (Turowski et al., 2009). A likely reason for this tendency is the high availability of fresh and unstructured sediment within the channel after large-magnitude floods (Brenna and Surian, 2023; Haddadchi and Hicks, 2019). A similar finding as for the Erlenbach field data was obtained by An et al. (2021) who used flume experiments to examine the effect of antecedent conditioning flows. They concluded that the effect of stress history on the sediment transport rate was limited to a relatively short time at the beginning of the hydrograph and decreased with

increasing flow and sediment supply.

The density distribution of $Q$ values for EDH and EDL conditions indicates that there is a separating value of about $Q = 0.645$ $m^3$ $s^{-1}$, below which EDH conditions occurred more frequently, and above which EDL conditions occurred more frequently, both for period A and B (Fig. 15a). If we consider the density distribution of $Q_{bM}$ values, the two distributions for EDH and EDL conditions are more similar (Fig. 15b); nevertheless, a separating value of about $Q_{bM} = 0.50$ kg $s^{-1}$ can be

determined, taken as the mean of the sum of the two median values of $Q_{bM}$ for periods A and B. This latter separating value of $Q_{bM} = 0.50$ kg $s^{-1}$ characterises typical transport conditions at the former separating value of $Q = 0.645$ $m^3$ $s^{-1}$. The separating value of a discharge of about 0.6 $m^3$ $s^{-1}$ (Fig. 15a) roughly corresponds to the transition between phase 1 and phase 2 transport conditions, which can be distinguished both in Fig. S9 and in Fig. 2 (for period B). A distinction between phase 1 and phase 2 transport conditions could also be observed with SPG measurements in two Austrian mountain streams

(Rickenmann, 2018).

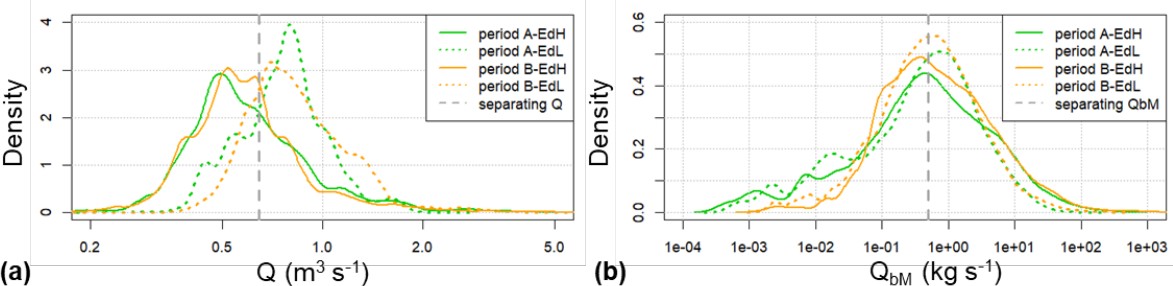

**Figure 15. Density distribution of two variables for EDH and EDL conditions for both periods A and B. (a) Density of $Q$ values, and (b) density of $Q_{bM}$ values. The vertical dashed red line indicates a separating value of $Q = $ ca 0.645 $m^3$ $s^{-1}$ (visually determined), above which EDL conditions and below which EDH condition dominate. A corresponding separating value is $Q_{bM} = $ ca 0.50 kg $s^{-1}$ (determined as mean of the median of the two periods).**





## 5 Conclusions

Fluctuations of bedload transport in the Swiss Erlenbach stream were analysed as a function of flow and transport conditions, using measurements with the Swiss Plate Geophone system with a temporal resolution of 1 min. This study
confirmed findings from an earlier event-based analysis of the same bedload transport data, which showed that the disequilibrium ratio (of measured to calculated transport rate) strongly influences the sediment transport behaviour. This study found further evidence that above average disequilibrium conditions, which are associated with a larger sediment availability on the streambed, generally have a stronger effect on subsequent transport conditions than below average disequilibrium conditions, which are associated with comparatively less sediment availability on the streambed.
First, the correlation between hydraulic forcing and bedload transport rate increased with increasing aggregation time and with increasing disequilibrium ratio. Second, a larger sediment availability on the bed increased the memory effect for bedload transport rates and for the disequilibrium ratio. Third, bedload transport fluctuations were clearly smaller during the rising limb of the hydrograph than during the falling limb. Fourth, bedload transport fluctuations appeared to be influenced by the sediment availability on the bed for the smallest discharge range (below 0.6 $m^3$ $s^{-1}$) and for the largest discharge range
(above 0.9 $m^3$ $s^{-1}$). Fifth, the dominant clockwise hysteresis was reduced for flood events with high shear stresses as well as for preceding large-magnitude floods that destabilised the channel bed.

## Appendix A1. Details of hydraulic and bedload-transport calculations

The lowermost channel reach with a natural bed upstream of the sediment retention basin has a trapezoidal cross-section, including partially engineered banks protected by a riprap construction (Fig. 1c). Several cross-sections were surveyed with a
total station; the average bottom width is 4.1 m and the banks have a lateral slope of 1.5:1 (Fig. 1c). The hydraulic calculations were carried out with an equation given in Nitsche et al. (2012, Fig. 5d therein) using the two dimensionless variables $U^{**}$ and $q^{**}$ for the mean flow velocity $U$ and the unit discharge $q$, respectively:

$$U^{**} = 1.49 \, q^{**0.6} \tag{A1}$$

$$U^{**} = U / \sqrt{gSD_{84}} \tag{A2}$$

$$q^{**} = q / \sqrt{gSD_{84}^3} \tag{A3}$$

where $g$ is the gravitational acceleration, $S$ is the channel bed slope, and $D_{84}$ is the grain size of the bed surface for which 84% of the particles are finer. Equation (A1) is based on dye tracer measurements made in the lowermost reach of the Erlenbach. The unit discharge $q$ was determined for a mean width for a given flow depth in the trapezoidal cross-section, and



bank resistance was accounted for by reducing $q$ with the ratio of the hydraulic radius $r_h$ to the flow depth $h$. This required an

iterative calculation procedure, using the measured discharge $Q_o$ along with the trapezoidal cross-section.

Bedload-transport calculations were performed with two equations given in Schneider et al. (2015), which represent a modified form of the Wilcock and Crowe (2003) equation. The first one (SEA1) is based on the use of total shear stress and a slope-dependent reference shear stress:

$$W_{tot}^* = 0.002 \left( \frac{\tau_{D50}^*}{\tau_{rD50}^*} \right)^{16.1} \qquad\qquad for \ \frac{\tau_{D50}^*}{\tau_{rD50}^*} < 1.143 \ and \ D > 4 \ mm \tag{A4a}$$

$$W_{tot}^* = 14 \left( 1 - \frac{0.85}{(\tau_{D50}^*/\tau_{rD50}^*)^{0.7}} \right)^{4.5} \qquad for \ \frac{\tau_{D50}^*}{\tau_{rD50}^*} \geq 1.143 \ and \ D > 4 \ mm \tag{A4b}$$

where the dimensionless transport rate $W^*_{tot}$ is defined as:

$$W^*_{tot} = Rgq_b/u^{*3} \tag{A5}$$

where $R = \rho_s/\rho - 1$ is the relative sediment density (with $\rho_s$ = sediment density and $\rho$ = water density), $q_b$ is the volumetric bedload transport rate per unit width, $u^* = (\tau/\rho)^{0.5}$ is the shear velocity, and $\tau = g\rho r_h S$ is the bed shear stress. $\tau^*_{D50} =$

$r_h S/(RD_{50})$ is the dimensionless bed shear stress with regard to the characteristic grain size $D_{50}$, and $\tau^*_{rD50}$ is the dimensionless reference bed shear stress with regard to the characteristic grain size $D_{50}$ of the bed surface. Equation (A4) was developed for calculation of total transport rates (not fractional transport rates). The threshold between low and high intensity transport in Eq. (A4) has been corrected to $\tau^*_{D50}/\tau^*_{rD50} = 1.143$, as compared to the value of 1.2 given in Schneider et al. (2015). $\tau^*_{rD50}$ is calculated as a function of the bed slope (Schneider et al., 2015, eq. 10 therein):

$$\tau^*_{rD50} = 0.56 \ S^{0.5} = 0.18 \tag{A6}$$

The bedload transport rate over the entire channel width, calculated with the total shear stress, is given as:

$$Q_{btot} = b_w \ W^*_{tot} \ u^{*3}/(Rg) \tag{A7}$$

The second equation (SEA2) is based on a reduced (effective) shear stress $\tau'$, using a reduced energy slope $S'$ (Rickenmann

and Recking, 2011; Rickenmann, 2012):

$$\tau' = g \ \rho \ r_h \ S' \tag{A8}$$

$$S' = S \ (f_o/f_{tot})^{0.5 \ e} \tag{A9}$$

where e = 1.5, $f_{tot}$ = friction factor for total flow resistance, calculated with eq. (A1), and $f_o$ = friction factor associated with grain resistance, calculated as (Rickenmann and Recking, 2011):

$$f_o = (8/6.5^2) \ (D_{84}/r_h)^{0.334} \tag{A10}$$



Then a constant, slope-independent dimensionless reference shear stress $\tau^*{}_{rD50} = 0.03$ is used to determine dimensionless transport rate $W^*{}_{red}$:

$$W^*_{red} = 0.002 \left(\frac{\tau^{*\prime}_{D50}}{\tau^{*\prime}_{rD50}}\right)^{7.8} \qquad for\ \frac{\tau^{*\prime}_{D50}}{\tau^{*\prime}_{rD50}} < 1.33\ and\ D > 4\ mm \tag{A11a}$$

$$W^*_{red} = 14 \left(1 - \frac{0.894}{(\tau^{*\prime}_{D50}/\tau^{*\prime}_{rD50})^{0.5}}\right)^{4.5} \qquad for\ \frac{\tau^{*\prime}_{D50}}{\tau^{*\prime}_{rD50}} \geq 1.33\ and\ D > 4\ mm \tag{A11b}$$


The bedload transport rate over the entire channel width, calculated with the reduced shear stress, includes the shear velocity $u^{*\prime} = (\tau^\prime/\rho)^{0.5}$ and is given as:

$$Q_{bred} = b_w\ W^*_{red}\ (u^{*\prime})^3/(Rg) \tag{A12}$$

For the development of Eq. (A4) and Eq. (A11), Schneider et al. (2015) used bedload transport measurements from 14 mountain streams, including samples from the Erlenbach that had been obtained with the moving basket system (Rickenmann et al., 2012).

## Appendix A2. Calculation of the hysteresis index HI_log following Lloyd et al. (2016)

The calculation of the hysteresis index $HI\_log$ follows the procedure developed by Lloyd et al. (2016), which is also reported
in the studies of Misset et al. (2018) and Vale and Dymond (2019). For the analysis of the bedload transport hysteresis in the Erlenbach, a slight modification was made by using logarithmic (instead of linear) values of the impulse counts ($IMP$) in calculating differences and normalizing these values (Eq. A15b). The hysteresis index $HI\_log$ was calculated as follows:

$$HI\_log = IMP_{RL\_norm} - IMP_{FL\_norm} \tag{A14}$$

where $IMP_{RL\_norm}$ is the normalised $IMP$ on the rising limb, and $IMP_{FL\_norm}$ is the normalised $IMP$ on the falling limb.
Normalisation of discharge ($Q$) differences and of $IMP$ differences was calculated for each time step i as:

$$Normalised\ Q_i = (Q_i - Q_{min}) / (Q_{max} - Q_{min}) \tag{A15a}$$

$$Normalised\ IMP_i = (\log(IMP_i) - \log(IMP_{min})) / (\log(IMP_{max}) - \log(IMP_{min})) \tag{A15b}$$

where $Q_i$ and $IMP_i$ represent discharge and impulse counts at a given timestep, $Q_{max}$ and $Q_{min}$ represent maximum and minimum discharge for a given flood event, and $IMP_{max}$ and $IMP_{min}$ represent maximum and minimum $IMP$ for a given flood
event. The spatial area contained within the hysteresis loop was calculated using the "polyarea" function within the package "pracma" in R Studio (RStudio Team, 2022). The hysteresis index Eq. (A14) calculates the difference of bedload transport



intensities on the rising and falling limbs and normalises the differences at every measurement point. This results in an index between -1 and 1, that is equal to 0 if there is no loop.

**Data availability**

The datasets presented in Fig. 2 form the basis of the analysis for this paper, and they are available online on the EnviDat repository (upon final publication): https://www.envidat.ch/#/metadata/sediment-transport-observations-in-swiss-mountain-streams (Rickenmann et al., 2023). Complementary event-based data, as described in Rickenmann (2020), is also available on the same repository.

**Competing interests**

The author declares that he has no conflict of interest.

**Acknowledgements**

I acknowledge the support of many colleagues at WSL involved in setting up and running the bedload transport measurements at the Erlenbach. I thank Alexandre Badoux of WSL who helped correcting an earlier version of the manuscript.

**List of variables**

| $ACF$ | autocorrelation coefficient |
|---|---|
| $b_w$ | bottom width of natural channel reach upstream of gauging station |
| $cv\_Q_{bM}$ | coefficient of variation of $Q_{bM}$ values |
| $D_{xx}$ | grain size of the bed surface for which xx% of the particles are finer |
| $E_d$ | disequilibrium ratio used in Rickenmann (2020), calculated as $M_{gravel}/M_{greg}$, where $M_{gravel}$ is the transported bedload mass per event, and $M_{greg}$ is the estimated bedload mass per event based on the calculation of $Q_{btot}$ |
| $E_{dks}$ | time-smoothened $E_{dM}$ values (see section 2.6 for details) |
| $E_{dM}$ | disequilibrium ratio, determined as $Q_{bM}/Q_{btot}$ (Eq. 2) |
| $E_{dM\_geo-mean}$ | geometric mean of disequilibrium ratio $E_{dM}$ |
| EDH | 50% of minutes values for which $E_{dks} >$ median($E_{dks}$), i.e. "high" $E_{dM}$ values |
| EDL | 50% of minutes values for which $E_{dks} <$ median($E_{dks}$), i.e. "low" $E_{dM}$ values |
| $f_g$ | lumped factor ($f_g$ = 22.46) in Eq. (1) |
| $HI\_log$ | hysteresis index (Appendix A2) |
| $IMP$ | impulse counts per minute (using the SPG system) |
| $k_{isp}$ | calibration coefficients determined for each survey interval |
| $LE_{dM\_mean}$ | Mean of log ($E_{dM}$) values |
| $LQ_{bM\_mean}$ | Mean of log ($Q_{bM}$) values |



| $LQ_{bt\_mean}$ | Mean of log ($Q_{btot}$) values |
|---|---|
| $M_{gravel}$ | bedload mass transported per event |
| $Q$ | water discharge (used in the analysis) |
| $Q_b$ | measured bedload transport rate (Eq. 1) |
| $Q_{bM}$ | measured bedload transport rate, zero values replaced by mean with neighboring non-zero values |
| $Q_{bM\_sum}$ | binned means of $Q_{bM}$, multiplied with 60 s and summed over the number of values per bin (i.e., a measured bedload mass) |
| $Q_{bred}$ | calculated bedload transport rate, based on the reduced (effective) shear stress (Appendix A1) |
| $Q_{btot}$ | calculated bedload transport rate, based on the total shear stress (Appendix A1) |
| $Q_{btot\_sum}$ | binned means of $Q_{btot}$, multiplied with 60 s and summed over the number of values per bin (i.e. a calculated bedload mass, reflecting flow intensity) |
| $Q_{bz}$ | measured bedload transport rate, including zero values |
| $Q_{bz\_sum}$ | binned means of $Q_{bz}$, multiplied with 60 s and summed over the number of values per bin (i.e., a calculated bedload mass, reflecting flow intensity) |
| $Q_e$ | discharge threshold at the end of an event |
| $Q_{max}$ | peak discharge of an event |
| $Q_{max\_p}$ | peak discharge of the preceding event |
| $Q_o$ | water discharge measured at the upper flow gauging station |
| $Q_s$ | discharge threshold at the start of an event |
| $Q_u$ | water discharge measured at the lower flow gauging station |
| $R$ | Pearson correlation coefficient |
| $S$ | channel slope of natural channel-reach upstream of gauging station (Appendix A1) |
| $S'$ | reduced energy slope (associated with the reduced shear stress) (Appendix A1) |
| $U$ | mean flow velocity (Appendix A1) |

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
