# Peer review of "Bedload transport fluctuations, flow conditions and disequilibrium ratio at the Swiss Erlenbach stream: results from 27 years of highresolution temporal measurements"

_EGUsphere, 2023_

## Author Comment (AC1)

**Response to the comments made by Anonymous Referee #1**

Dear Referee #1,

I thank you for your positive assessment of the manuscript. I appreciate your valuable comments that have helped to improve the manuscript. I agree with most of your suggestions, and I have revised the manuscript accordingly. Below, the reviewer comments are reported in italics, and my responses in normal font (blue colour).

*In this paper, the author uses 27 years of Swiss plate geophone recordings in the Erlenbach to analyze the dynamics of bedload transport in mountain streams. The paper considers an original protocol that compares measured transport with capacity transport. The results show that transport intensity and fluctuations are strongly correlated with pre-flood conditions, and more specifically with sediment availability in the river bed.*

*I found the article well-written with an exhaustive review of the literature, but also very difficult to read, mainly due to the excessive number of parameters and abbreviations (e.g. 15 flow parameters Q_x with different x-indices). But I found the methodology interesting and think that the proposed framework may be useful for future work.*

Response:     I agree that many different flow parameters were defined and used. For this reason, I have included a list of variables with the definitions of the abbreviations and indices. In order not to impair the transparency of the analysis, I think that it is difficult to reduce the number of variables. (In the case of $Q_\text{o}$ and $Q_\text{u}$ a slight simplification has already been made, after the introduction and explanation of the discharge measurements in section 2.3, by using only this variable $Q$ in the following text.)

*I have no reservations about publication with minor revisions.*

Comments

*Line 158: If I've understood correctly, the signal emitted by the >10mm fraction is used as a proxy to estimate the transport rate associated with the 4mm to 10mmfraction, by calibration? Perhaps this could be written more explicitly.*

Response:     No, the SPG signal (which is sensitive to particles with $D > 10$ mm [L 138]) is used to estimate the fraction with $D > 10$ mm. The fraction with $4\text{mm} < D < 10$ mm is accounted for by introducing the constant factor $m_\text{fg} = 1.54$, as explained in L164-170.

*Line 174: does "zero values" means periods with no impulses in the time series?*

Response:     Yes. There are essentially two types of time periods with $Q_\text{b}$ values of zero. (i) Longer periods between events with insufficient discharge for bedload transport (as detected

by the SPG system) due to no or insufficient rainfall (or snowmelt). (ii) Shorter periods of tens of minutes to a few hours, after which an increase in discharge may reactivate bedload transport. In the present study, I used the time series of the 522 flood events as defined in Rickenmann (2020), which effectively excluded zero $Q_b$ values of type (i). Following your comment, I have now included the information on the time series used in this study.

**Line 215**: *it is not trivial to apply a bedload equation in such a torrent context. Could you specify which part of the upstream reach and which grain size distribution were considered for the calculations? (in particular I suppose grain size varies a great deal through time?)*

Response:      The reach location and the grain size information were already described in L216-219. I have added now more information on the length of the reach and the likely variability of grain size distribution.

**Line 253**: *is there any reason why you use Qbtot and not Qbred in Eq2?*

Response:      The reasons are that I had also used $Q_{btot}$ in the event-based analysis in Rickenmann (2020) and that $Q_{btot}$ better approximates the mean trend of the measured transport rates. This explanation has been added to the manuscript.

**Line 257**: *what are Qb,s and Qb,e?*

Response:      These are old notations (erroneously retained), which have been changed in most locations to $Q_s$ and $Q_e$ (for easier readability), and which have been corrected in the revised version of the manuscript also in this particular case.

**Line 286**: *correct "in the in the"*

Response:      corrected

**Line 286-9**: *it means that larger Q are associated with smaller Qbm?*

Response:      Yes. The main purpose of this part of the analysis was to compare general transport characteristics between periods A and B. From Text S1 and Figures S2-S8 it can be inferred that for $Q < 1$ m$^3$/s there are larger $Q$ values associated with smaller $Q_{bm}$ values in period A than in period B (e.g. Figure S6). Essentially, for smaller discharges, there are relatively more zero and smaller $Q_b$ values in period A (than B), whereas for larger discharges there are smaller $Q_b$ values in period A (than B).

**Line 283**: *the analysis presented below does not really explain why the wider spread in period B. Or maybe I missed an information?*

Response:      At this location, no (hypothetical) explanation is given. A possible reason for the wider spread in period B than A is likely linked to more prevalent "phase 1" transport conditions (L326-330, old manuscript) in period B than A. Such conditions are assumed to result from a generally higher sediment availability in period B than A (L477-479, old manuscript).

**Line 317**: *How did you delimitate these periods? Visually?*

Response:      Yes, first visually, and then the exact limits were determined based on the data to match the end and start of the flood events (i.e. flood-event limits, L318). This information has been added to the manuscript.

*Line 346: why 30 min?*

Response:      The 30 min time frame was selected, because there is a relatively strong increase in the correlation between $Q_b$ and $Q$ for aggregation times up to 30 min (Figure 4), and because most event durations are considerably longer than 30 min (Figure S1). This information has been added.

*Line 354: is "log values of EdM (ACF_LEdM_m30)" correct?*

Response:      Yes, this is correct.

*Fig7: I guess it is an important result, but it is still difficult for me to appreciate the physical meaning of the relation between Qb and the autocorrelation coefficient…*

Response:      The autocorrelation of bedload transport rates within a time window of 30 min in this study is likely to be partly associated with collective entrainment of particles (e.g. Ancey, 2020a; Ma et al., 2014), as compared to completely random fluctuations that would reflect white noise behaviour without autocorrelation. This information has been added to the Discussion at the end of section 4.1.

*Line 361: how did you define (measure) the critical discharge?*

Response:      Critical discharge was determined by the start and end of bedload transport activity, as determined from the SPG signal, in combination with the delineation of sediment-transporting flood events (Rickenmann, 2020). The partly missing information in the manuscript has been added to section 2.4.

*Line 263: it may be useful to recall here the definition "discharge threshold at the end of an event". Same for Qs.*

Response:      I disagree here, because repeating definitions of notation symbols would lengthen the text unnecessarily (for which symbols would a repeated definition be necessary, and for which not?).

*Fig9 and 10: why do we observe a positive correlation of Qe with QbT and a negative correlation with QbM? I missed something…*

Response:      Here it is important to recall that $Q_{btot}$ is a measure of the hydraulic forcing and $Q_{bM}$ is the observed transport rate, and further that the disequilibrium rate is defined as $E_{dm} = Q_{bM} / Q_{btot}$. Strong negative correlations were observed between $Q_s$ or $Q_e$ and $E_{dm}$ (Figure 8, and event-based analysis in Rickenmann, 2020). Given this observation, it is reasonable to find a positive correlation between $Q_e$ and $LQ_{bt}$, and a negative correlation between $Q_e$ and $LQ_{bM}$.

*Fig10: what about the correlation between Qs and LQbt?*

Response:      First, there was a error in Figure 10a: the label of the abscissa should be $Q_{bM\_mean}$, i.e. the linear mean of the $Q_{bM}$ values (this has been corrected), as opposed to the log mean ($LQ_{bM\_mean}$) in Figure 10b. The main point to illustrate here is that the correlation between $Q_e$ and $Q_{bM\_mean}$ is moderate (Figure 10a, corrected x-label) and vanishes if $LQ_{bM\_mean}$ values are used (Figure 10b). the correlation between $Q_s$ and $Q_{bM\_mean}$ is weaker and not shown.

**Lines 383-386**: *brackets are not necessary*

Response:      corrected

**Line 483**: *because more in-channel material immediately available for transport?*

Response:      Yes, this likely reason has been added to the revised text.

**Line 443**: *Phase 1 has been mentioned in different places for explaining the data in period B. I think it could be interesting to recall the exact definition of phase 1 used in this paper and also to indicate the t\*/tc\* range concerned?*

Response:      Phase 1 transport conditions are defined in L328, so I prefer not to repeat the definition here. Instead, I have added the ref. (Ryan et al., 2002). It makes sense to indicate the range concerned, but for better illustration this range is given in terms of discharge ("for $Q$ values smaller than about 0.5 m$^3$/s"), to be compatible with Figure S9 that is referenced in the main text.

**Line 481**: *this sentence is not clear*

Response:      I agree that this sentence is difficult to understand. The intention was to explain partly the very large (relative) fluctuations of $Q_b$ in this discharge range (as compared to larger discharges). However, this point is not a special focus of the paper, so for the sake of clarity I decided to omit this sentence in the revised version of the manuscript.

**Line 489-490**: *Your work conclude that the bed plays important role in what was measured. We can regret that you don't propose a paragraph dedicated to your direct bed observations (if any?). and do you have indications on the size of the transported materials for both periods?*

Response:      Thank you for this suggestion. I agree that the issue of sediment availability on the streambed is central to the discussion of the findings of this study. Therefore, a paragraph on streambed sediment availability  has been added at the end of section 2 in the revised text.

**Further changes**

I have also made some further minor changes to the original manuscript. These are mainly typos. All changes can be found in the "tracked-changes" version of the revised manuscript.

---

## Author Comment (AC2)

**Response to the comments made by Anonymous Referee #2**

Dear Referee #2,

I thank you for your critical assessment of the manuscript. I appreciate your valuable comments that have helped to improve the manuscript. I agree with most of your suggestions, and I have revised the manuscript accordingly. Below, the reviewer comments are reported in italics, and my responses in normal font (blue colour).

*This work presents an analysis of an extensive set of geophone and piezoelectric sensor data that are used to infer bed sediment transport in an Alpine stream. The primary focus of the paper the collection and processing of these particular data, and correlations in the results.*

*The paper is closely related to the work of Rickenmann (2020), which presented an analysis of the same data, but based on events rather than per-minute observations. Although it is reassuring to see that the change in methodology does not significantly change the results, in my view a shortcoming of the paper is the similarity to this previous work.*

*My main comment is therefore to suggest that a more direct comparison of the strengths and weaknesses of the two approaches is made, in order to demonstrate that the method presented here is indeed a substantial advance. In particular, the scientific benefit would be demonstrated by showing that the new methodology has the ability to test specific scientific hypotheses that could [NOT] be tested with the previous approach.*

Response:    I assume that the last statement in the third paragraph should rather read: "that could NOT be tested with the previous approach." I would like to mention here first that the similarity of the analyses only concerns a part of this study, i.e. mainly the results presented in Figures 6 to 10. However, in the earlier study (Rickenmann, 2020) all flood events had the same weight in the analysis (independent of the event duration), whereas in this study each single observation on bedload transport (i.e., 1 min value) had the same weight. All the other results, presented in Figure 4 and Figures 11 to 15, are completely new and could not have been obtained using the event-based analysis presented in Rickenmann (2020). Regarding the first part of the analysis in this study (i.e. the results related to Figures 6 to 10), I will use the expression "minute-based" analysis in the following, to distinguish it from the earlier event-based analysis. In fact, in the first part of this study, the minute-based analysis examined longer time intervals than the event-based analysis, whereas in the second part of this study, the minute-based analysis considered shorter time intervals than the event-based analysis, and also examined variations in the coefficient of variation of the transport rate and hysteresis effects. To make these differences clearer, I have introduced a short paragraph at the beginning of the discussion section.

I would also like to mention two important elements that may have influenced the results of the two types of analysis in different ways: (i) Between the sediment-transporting flood events in the Erlenbach, streambed characteristics may change due to sediment supply from the hillslopes, as discussed in Turowski et al. (2011). They showed that $Q_s$ of a given event can be different from $Q_e$ of the previous event, partly due to this phenomenon; (ii) between the

sediment-transporting flood events in the Erlenbach, an armouring effect on the streambed can also occur due to below-threshold flows (with no bedload transport according to the SPG measurements). Such flows are still sufficiently high to cause a rearrangement of the particles on the bed, as discussed in Masteller et al. (2019). Now, in general it may be concluded that the results of the minute-based analysis confirmed the results of the event-based analysis.

In both types of analysis, the effects of the elements (i) and (ii) were not considered explicitly. If they had been (very) important, they might have affected the results of the minute-based analysis more strongly, because in this analysis time intervals were used (i.e. periods p1 through p13) that each contained several events. From this assessment and given the general similarity of the results from the two types of analyses, it may be hypothesized that the effect of (variable) sediment availability on the streambed was more important than an effect of elements (i) and (ii) on the results. This concerns the results regarding the autocorrelation of bedload transport rates and disequilibrium ratio (Figure 6, 7) and the correlations between threshold discharges ($Q_s$, $Q_e$) and either disequilibrium ratio ($E_{dM}$, Figure 8) or hydraulic forcing ($Q_{btot}$, Figure 9).

*I concur with anonymous referee #1's list of comments, and suggest a few more minor clarifications:*

**Abstract**: *a certain amount of jargon is used here (disequilibrium ratio, lag time, critical discharge, coefficient of variation, clockwise/anticlockwise transport behaviour) much of which is likely to be unclear to people who have not already read the paper.*

Response:     I have changed some expressions in the abstract to make it easier to read. However, the use of some technical terms has been retained in favour of a more direct connection with the main text.

**line 78**: *define coefficient of variation*

Response:     An explanation has been added to the revised manuscript.

**line 219**: *"xx% of the particles are finer" presumably refers to particle mass, rather than particle number?*

Response:     This has been specified in the revised manuscript.

**line 262**: *Clarify exactly what the 'kernel smoothing' does (presumably a type of low-pass filter?) and what the unit of bandwidth is? (I'd usually understand bandwidth to be measured in Hz).*

Response:     Yes, the kernel smoothing is used here as a type of low-pass filter. The bandwidth defines the number of neighbouring points that are included in the smoothing window. The selected bandwidth of 30 with a Gaussian kernel (used here) smoothens over a window of roughly 60 minute-values. This number was selected because it resulted in a smoothing of the short-time fluctuations of bedload transport (with an associated increase in the correlation between $Q_b$ and $Q$, Figure 4) and because the majority of the events have longer durations. This information has been added at the end of section 2.6.

**Further changes**

I have also made some further minor changes to the original manuscript. These are mainly typos. All changes can be found in the "tracked-changes" version of the revised manuscript.